# The dynamic lateral gate of the mitochondrial β-barrel biogenesis machinery is blocked by darobactin A

Kathryn A. Diederichs [1], Istvan Botos [1], Scout Hayashi [1], Gvantsa Gutishvili [2], Vadim Kotov[3,4,5], Katie Kuo [2], Akira Iinishi[6], Gwendolyn Cooper [7], Benjamin Schwarz [7], Herve Celia [1], Thomas C. Marlovits [3,4,5], Kim Lewis[6], James C. Gumbart [2], Joseph A. Mindell [8] ✉ & Susan K. Buchanan [1] ✉

The folding and insertion of β-barrel proteins into the mitochondrial outer membrane is facilitated by the sorting and assembly machinery (SAM) complex. Here we report two 2.8 Å cryo-EM structures of the *Thermothelomyces thermophilus* SAM complex in the absence of substrate in which the Sam50 lateral gate adopts two different conformations: the first is a closed lateral gate as observed in previously published structures, while the second contains a Sam50 with the first four β-strands rotated outwards by approximately 45°, resulting in an open lateral gate. The observed monomeric open conformation contrasts our previous work where the open conformation was adopted by non-physiological up-down dimers. To understand how these lateral gate dynamics are influenced by substrate, we studied the interaction of the SAM complex with a β-signal peptide mimic, darobactin A. Darobactin A binds to the SAM complex with nanomolar affinity and inhibits the import and assembly of mitochondrial β-barrel proteins in vitro. Lastly, we solved a 3.0 Å cryo-EM structure of the *Thermothelomyces thermophilus* SAM complex bound to darobactin A, which reveals that darobactin A stabilizes the Sam50 lateral gate similar to the open conformation by binding to strand β1, therefore blocking β-barrel biogenesis.

The outer membranes of mitochondria and Gram-negative bacteria contain integral membrane proteins with a β-barrel topology that perform a wide variety of essential functions. The outer mitochondrial membrane (OMM) of yeast contains four types of β-barrel proteins (Sam50, Tom40, Mdm10, and VDAC) which contain 16 or 19 β-strands[1,2]. In contrast, the outer membrane of Gram-negative bacteria contains a wider variety of β-barrel proteins, which can have anywhere from 8-36 β-strands[3,4]. Despite these differences, all β-barrel proteins start by being translated in the cytosol, translocated to the intermembrane space (mitochondria) or periplasm (Gram-negative

[1]Laboratory of Molecular Biology, National Institute of Diabetes & Digestive & Kidney Diseases, National Institutes of Health, Bethesda, MD, USA. [2]School of Physics, Georgia Institute of Technology, Atlanta, GA, USA. [3]University Medical Center Hamburg-Eppendorf (UKE), Institute for Microbial and Molecular Sciences, Hamburg, Germany. [4]Centre for Structural Systems Biology (CSSB), Hamburg, Germany. [5]Deutsches Elektronen Synchrotron (DESY), Hamburg, Germany. [6]Antimicrobial Discovery Center, Department of Biology, Northeastern University, Boston, MA, USA. [7]Proteins & Chemistry Section, Research Technologies Branch, Division of Intramural Research, National Institute of Allergy and Infectious Diseases, National Institutes of Health, Hamilton, MT, USA. [8]Membrane Transport Biophysics Section, National Institute of Neurological Disorders and Stroke, National Institutes of Health, Bethesda, MD, USA. ✉e-mail: mindellj@ninds.nih.gov; susan.buchanan2@nih.gov

bacteria) as unfolded precursors, bound by chaperones, and eventually assembled into the outer membrane (Supplementary Fig. 1)[1,5].

The folding and insertion of β-barrel proteins into the outer membrane is facilitated by evolutionarily conserved machineries[6]; the sorting and assembly machinery (SAM) complex in mitochondria[6–9], and the β-barrel assembly machinery (BAM) complex in Gram-negative bacteria[10,11]. The SAM complex is composed of three subunits: β-barrel Sam50 embedded in the OMM, and two cytosolically associated subunits Sam35 and Sam37 (Supplementary Fig. 1A)[12–19]. In contrast, the BAM complex from *E. coli* has five subunits: membrane-embedded β-barrel BamA, and lipoproteins BamB-E which interact with the periplasmic side of BamA and the outer membrane (Supplementary Fig. 1B)[10,11,20,21].

Sam50 and BamA are essential for β-barrel biogenesis[12,14,22–25], and share structural topology of a 16-stranded β-barrel with N-terminal polypeptide transport associated (POTRA) domain(s) extending into the intermembrane space (Sam50) or periplasm (BamA)[17–19,26–32] as well as sequence similarity (Supplementary Fig. 2). The seam where the first and last β-strands of the Sam50 or BamA β-barrel meet is called the lateral gate. Unlike other β-barrel proteins in which the seam stays statically closed, the lateral gate of Sam50 and BamA is flexible and required to open for function[9,33,34]. Despite the similarities in topology and sequences of Sam50 and BamA, all Sam50 structures solved to date in the absence of substrate contain a lateral gate with no hydrogen bonds between the first and last β-strands[17–19]. In contrast, BamA structures in the absence of substrate contain at least one hydrogen bond between β1 and β16 stabilizing a closed lateral gate[26,32,35].

Despite the similarities in core components, Sam50 and BamA, the accessory proteins of the SAM and BAM complexes do not share any sequence or topology similarities. In fact, the accessory subunits interact with Sam50/BamA from opposite sides of the membrane, suggesting they may have distinct functions in each system[17–19,27–31,34,36].

The SAM and BAM complexes recognize substrate β-barrel proteins by a conserved sequence motif on the most C-terminal β-strand, called the β-signal[12,20,37,38]. These sequences are similar enough that bacterial β-barrel proteins can be recognized and folded by the SAM complex and mitochondrial β-barrel proteins can be recognized and folded by the BAM complex[39,40]. Crosslinking studies of Sam50 with substrate β-barrels demonstrated that the β-signal specifically interacts with β1 of Sam50, displacing β16[9]. This observation was further supported by a recent cryo-EM structure of the SAM complex in the process of folding Tom40 in which the Tom40 β-signal is interacting with Sam50 β1 (SAM$^{stall}$)[34]. Structural studies of the BAM complex bound to substrate also demonstrated that the β-signal binds β1 of BamA[41–43].

Recently, a naturally occurring cyclized peptide containing the β-signal sequence motif, darobactin A, was isolated from *Photorhabdus* and found to have antibiotic activity against Gram-negative bacteria by modulating BAM complex activity[44–47]. Darobactin A interacts with BamA β1[47], blocking association of native substrate β-signals[46], and ultimately leading to cell death[44]. Interestingly, a linearized form of darobactin A containing no backbone cyclizations (linear darobactin) did not interact with the BAM complex with a measurable affinity[47], suggesting that a restricted secondary structure conformation is required for stable interaction with the BAM complex.

We set out to improve understanding of the SAM complex lateral gate dynamics and β-signal recognition by the SAM complex. By solving cryo-electron microscopy (cryo-EM) structures of the *T. thermophilus* SAM complex in the absence of substrate at 2.8 Å resolution, we found that Sam50 can adopt an open or closed lateral gate. Additionally, we determined that darobactin A binds to the SAM complex with nanomolar affinity and inhibits β-barrel biogenesis in vitro. Our 3.0 Å cryo-EM structure of the SAM complex bound to darobactin A shows that darobactin A binds to β1 of Sam50 and stabilizes an open conformation of the lateral gate.

## Results

### Purification of monomeric SAM complex

We previously purified the SAM complex from *Thermothelomyces thermophilus* in detergent and observed non-physiological up-down dimers of the SAM complex (two copies of each subunit) during structural characterization[19]. To study a more physiologically relevant form, we optimized purification conditions to promote monomeric SAM populations (complex of Sam50:Sam35:Sam37 in a 1:1:1 stoichiometry). Thermal unfolding screens and MoltenProt analysis[48], identified arginine as a buffer additive that increased the stability of the purified SAM complex (Supplementary Fig. 3A, B). Blue-native polyacrylamide gel electrophoresis (BN-PAGE) analysis revealed that the addition of 200 mM arginine in early purification steps promoted the SAM complex monomer population in LMNG/GDN (Supplementary Figs. 3C, D, 4A-C) and GDN (Supplementary Fig. 4G-I).

### Two conformations of SAM complex distinguished by cryo-EM

The monomeric SAM complex sample purified in GDN was then used for cryo-EM studies. Ultimately two different monomeric SAM complex structures could be distinguished at 2.8 Å resolution (Fig. 1A–E, Table 1, Supplementary Figs. 5, 6), from the same sample. Both structures contained clear density for Sam50, Sam35, and Sam37. The first structure contained Sam50 with a closed lateral gate (201,630 particles, Fig. 1A,C), now referred to as SAM$^{cl}$, as observed in previously published structures[17–19]. In contrast, the second structure contained Sam50 with the first four β-strands rotated outwards resulting in an open lateral gate (200,134 particles, Fig. 1B, D), now referred to as SAM$^{op}$. Prior to the structures presented here, an open Sam50 lateral gate had only been observed in non-physiological dimers or in a stalled folding intermediate[19,34].

The presence of both closed and open Sam50 lateral gate populations in our cryo-EM dataset suggests that the SAM complex samples both conformations in the absence of substrate (Supplementary Movie 1). Additionally, slight differences in the POTRA domain conformation are observed, with the SAM$^{cl}$ POTRA domain slightly shifted away from the Sam50 β-barrel lumen and the SAM$^{op}$ POTRA domain slightly shifted underneath the barrel lumen. The essential Sam50 cytosolic loop 6 (L6)[9] also did not undergo any large conformational changes between the SAM$^{cl}$ and SAM$^{op}$ structures (Fig. 5E), suggesting that this loop plays a role in stabilizing the rest of the Sam50 β-barrel during lateral gate opening. Although the structures presented here were solved with protein purified in detergent, the SAM$^{cl}$ structure presented here superimposes with the cryo-EM structure of the previously solved *Tt*SAM complex in lipid nanodiscs[19], with an overall RMSD of 1.06 Å.

The accessory subunits in the open and closed structures are highly similar (Fig. 1E) with Sam35 and Sam37 superimposing with C-alpha RMSDs of 0.33 Å and 0.47 Å, respectively (Supplementary Table 3). Additionally, there are several lipid densities in both maps on the back side of the Sam50 β-barrel (opposite the lateral gate) as well as beneath the Sam37 cytosolic domain (Fig. 1A, B). We have modeled these lipids as ergosterol molecules, as the size and density features of these regions are consistent with ergosterol. Additionally, LC/MS analysis of the SAM complex purified in GDN confirmed the presence of ergosterol (Supplementary Fig. 7). The lipid densities in both SAM$^{op}$ and SAM$^{cl}$ maps are at the same positions, suggesting that these are preferred binding sites for lipid. While ergosterol is present in the yeast OMM in relatively low amounts[49], it is possible that these molecules specifically and tightly bind to the SAM complex and remain bound during purification with GDN, a relatively mild detergent.

### MD simulations uncover Sam50 lateral gate dynamic features

To explore the dynamics of Sam50, we utilized high-resolution structures from *T. thermophilus* (PDB IDs: 6WUH, 6WUT) and *S. cerevisiae* (PDB IDs: 7BTX, 7E4H). This allowed us to construct four β-barrel-only

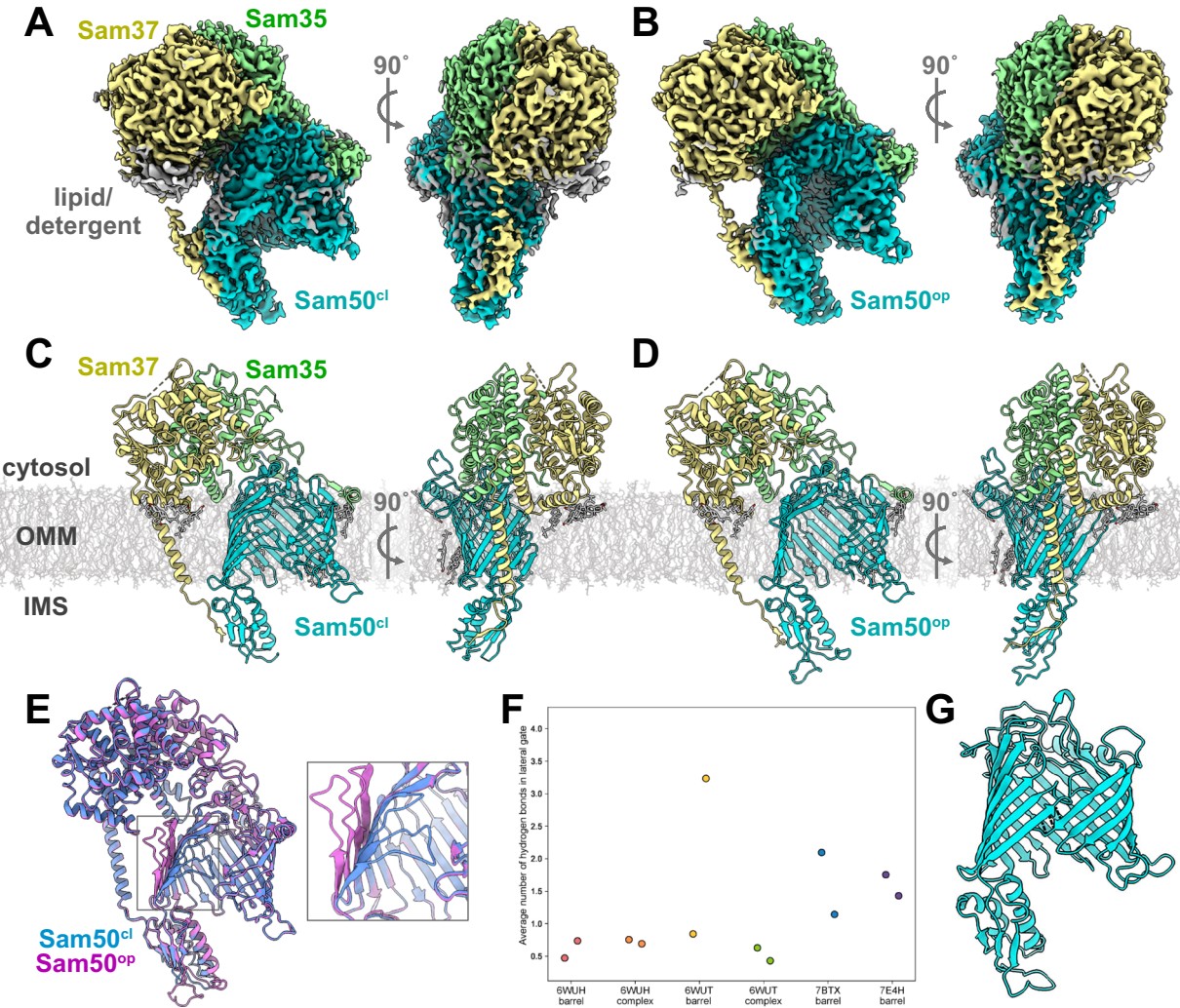

**Fig. 1 | Cryo-EM structures of the SAM complex purified in detergent reveal a dynamic lateral gate.** Cryo-EM maps of the *T. thermophilus* SAM complex colored by subunit (Sam50 in teal, Sam37 in yellow, Sam35 in green, density attributed to lipid or detergent molecules in gray). **A** SAM$^{cl}$ density front and side view. **B** SAM$^{op}$ density front and side view. (**C**) SAM$^{cl}$ structure front and side view. **D** SAM$^{op}$ structure front and side view. **E** Superposition of SAM$^{cl}$ (blue) and SAM$^{op}$ (pink) generated using ChimeraX v1.4 matchmaker[86,99]. Lateral gate region shown in zoom

inset. **F**–**G** MD simulations reveal a Sam50 β-barrel in a stable closed conformation. β1 and β16 are connected with, on average, three hydrogen bonds. **F** Average number of hydrogen bonds between the β1 and β16 strands of Sam50 for the last 2 μs of each simulation. Each data point represents a single replica. **G** Sam50 in a zipped closed conformation, observed with, at most, a couple of hydrogen bonds (black dotted lines). See also Supplementary Figs. 5, 6. Source data are provided as a Source Data file.

systems and two full SAM complexes. Each system was embedded in a native-like membrane environment and solvated with TIP3P water molecules and K$^+$ and Cl$^-$ ions (see "Methods"). We conducted two replicas of each simulation, each lasting 5 μs, resulting in a total simulation time of 60 μs. We observed a fully zipped closed lateral gate in only one replica of the 6WUT system, with up to four hydrogen bonds between the β1 and β16 strands (Fig. 1F, Supplemental Movie 2), while other simulations show a semi-closed, unstable lateral gate with 0–2 hydrogen bonds between the β1 and β16 strands (Fig. 1G). The zipped closed conformation has a few hydrogen bonds, which distinguishes it from the SAM$^{op}$ and SAM$^{cl}$ conformations that have no hydrogen bonds. Despite the lateral gate being at least partially open in most simulations, lipids were generally absent from the interior of the gate. In one replica of the *T. thermophilus* SAM complex (6WUH-complex), however, we observed a DOPC lipid becoming stuck within the lateral gate (Supplementary Movie 3); in the other replica, DOPC approached the gate but did not become lodged in it. These observations suggest that although lipids may approach the lateral gate, they do not penetrate the β-barrel.

Additionally, we monitored the membrane thickness over time, finding a distinct thinning near the lateral gate (Supplementary Fig. 8). While the regular average membrane thickness is approximately 30 Å, we observed a reduced thickness ranging from 13 Å to 25 Å near the lateral gate, depending on the specific system (Supplementary Fig. 8). This behavior is akin to observations previously reported for BamA[50].

## Darobactin A binds the SAM complex with nanomolar affinity

We next sought to advance mechanistic understanding of β-signal recognition by the SAM complex and interrogate the role of lateral gate dynamics in this early step of β-barrel biogenesis. Based on its interactions with BAM, we chose darobactin A, a naturally occurring cyclic peptide that contains a β-signal motif, as a substrate that might mimic the first step of β-signal binding from the β-barrel biogenesis pathway.

Using microscale thermophoresis (MST), we found that the SAM complex binds darobactin A with 296 nM affinity (Fig. 2A, C, E). Some variation in binding affinity (95.6−377 nM) was observed between

**Table 1 | Cryo-EM data collection, structure determination, and model statistics**

| | SAM$^{op}$ | SAM$^{cl}$ | SAM$^{daro}$ |
|---|---|---|---|
| *Data collection* | | | |
| Nominal magnification | 105,000x | | 105,000x |
| Voltage (kV) | 300 | | 300 |
| Exposure time (s/frame) | 0.05 | | 0.05 |
| Number of frames | 40 | | 50 |
| Total dose (e-/Å²) | 70.0 | | 54.4 |
| Defocus range (μm) | −0.6 to −1.6 | | −0.7 to −2.0 |
| Pixel size (Å) | 0.412/super-res pix | | 0.415/super-res pix |
| *Image processing* | | | |
| Micrographs selected | 13,205 | | 9235 |
| Initial particle images (no.) | 8,542,808 | | 3,909,861 |
| Final particle images (no.) | 200,134 | 201,630 | 163,391 |
| Symmetry imposed | C1 | C1 | C1 |
| FSC threshold | 0.143 | 0.143 | 0.143 |
| Final map resolution (Å) | 2.84 | 2.88 | 3.00 |
| *Atomic model* | | | |
| Number of protein residues | 1142 | 1121 | 1149 |
| *Validation* | | | |
| Favored (%) | 97.35 | 97.20 | 98.23 |
| Allowed (%) | 2.65 | 2.80 | 1.77 |
| Outliers (%) | 0.00 | 0.00 | 0.00 |
| Rotamer outliers (%) | 0.11 | 0.69 | 0.56 |
| r.m.s.d Bond lengths (Å) | 0.003 | 0.004 | 0.004 |
| r.m.s.d Bond angles (°) | 0.596 | 0.640 | 0.647 |
| Clashscore | 8.04 | 8.40 | 5.99 |
| Map CC | 0.87 | 0.88 | 0.87 |
| *Deposition ID* | | | |
| PDB | 9NK7 | 9NK6 | 9NK8 |
| EMDB | EMD-49495 | EMD-49494 | EMD-49496 |

biological replicates (Supplementary Table 1). This is similar to the binding affinity of the BamA β-barrel in detergent to darobactin A ($K_d$ = 0.6 μM) determined by ITC[47].

In contrast, a linear darobactin A peptide binds to the SAM complex with millimolar affinity ($K_d$ = 0.49–1.2 mM) (Fig. 2B, D, E, Supplementary Table 1). Because the linear darobactin binding curves do not saturate, the $K_d$ values reported represent the lower limit of $K_d$ and suggest that the shift between darobactin A and linear darobactin is at least three orders of magnitude. Weaker binding affinity was also observed for linear darobactin interactions with the BAM complex[47]. Linear β-signal peptides from mitochondrial β-barrels Porin1 or Tom40 (Supplementary Table 2) bound to the SAM complex in detergent with micromolar affinities (Fig. 2F, Supplementary Table 1), which is weaker than the SAM complex interaction with darobactin A but stronger than the linear darobactin interaction. The weaker binding affinities of linear darobactin and linear β-signal peptides suggest that the restricted secondary structure of darobactin A due to the backbone cyclizations contributes to stable association with the SAM complex. The stronger binding affinities of the linear β-signal peptides compared to linear darobactin may be a result of the peptide length. The β-signal peptides are 21 (Porin1) or 27 (Tom40) amino acids in length, whereas the linear darobactin peptide is only 7 amino acids long. The extended length of the β-signal peptides may allow them to adopt a β-hairpin conformation, while the linear darobactin peptide only contains enough residues to form a single β-strand. β-hairpin structures exhibit reduced

conformational flexibility compared to single β-strands due to hydrogen bonds stabilizing the two β-strands forming the β-hairpin.

## Darobactin A inhibits SAM complex function in vitro

We next asked if the binding of darobactin A to the SAM complex inhibits mitochondrial β-barrel assembly in vitro. To assess this, we used an in vitro mitochondrial import assay in which isolated *S. cerevisiae* mitochondria were incubated with darobactin A for four minutes before the addition of rabbit reticulocyte lysate containing radiolabeled precursor protein and further incubation to allow for SAM-induced protein folding. Following incubation, the mitochondrial outer membrane was solubilized with digitonin, and the soluble fraction was subjected to BN-PAGE and autoradiography. We focused on the import and assembly of the mitochondrial proteins relevant to the SAM complex function, β-barrel proteins, and did not evaluate the import of other mitochondrial proteins, such as pre-sequence containing proteins.

The first substrate we tested was Tom40, a β-barrel pore central to the translocase of the outer membrane (TOM) complex, as this is the slowest mitochondrial β-barrel to be released from the complex[6,8,51,52]. Additionally, the import and assembly of Tom40 has been well characterized and distinct steps have been identified with BN-PAGE[8,9,12,25,52–55]: [1] Tom40 association with the SAM complex and start of folding, [2] association of small TOM subunits with the Tom40 β-barrel and release of the intermediate complex, [3] dimerization of the intermediate complex and association of large receptor subunits to form the mature TOM complex. In the absence of darobactin A, we observe import and assembly of radiolabeled Tom40 over the course of 5–60 minutes following the previously observed folding and complex formation steps (Fig. 3A). Tom40 import and assembly were slightly diminished when 20.7 μM darobactin A was included in the import reaction, though a small amount of mature TOM complex was still obtained by 60 minutes. However, in the presence of 103.5 μM darobactin A TOM complex assembly was dramatically reduced with a very small amount of mature TOM complex formed after 60 minutes. These data indicate that the presence of darobactin A indeed inhibits SAM complex function in vitro.

Similarly, we observed robust import and assembly of the three other yeast mitochondrial β-barrel proteins (Mdm10, Porin1, and Sam50) in the absence of darobactin A over the course of 4–60 minutes (Fig. 3B–D). As with Tom40, import and assembly were slightly reduced in the presence of 20.7 μM darobactin A, and dramatically reduced with 103.5 μM. Porin1 assembly appears slightly less sensitive to darobactin A compared to other substrates, based on band intensity after 60 minutes in the 103.5 μM darobactin A sample (Fig. 3C). Given the high abundance of Porin1 relative to other mitochondrial outer membrane β-barrels[56], the observation of human VDAC1 (the human Porin1 homolog) to spontaneously fold into lipid bilayers in vitro[57], and the different impact of darobactin A on Porin1 assembly, it is possible that the folding mechanism of Porin1 is slightly different from other β-barrels, and/or Porin1 might use a few different folding mechanisms.

Our data demonstrate that darobactin A inhibits SAM complex function in vitro, which is a concern for the use of darobactin A as an antibiotic to specifically treat Gram-negative bacterial infections, but which does not address the compound's in vivo activity. Notably, darobactin A did not affect *S. cerevisiae* cell growth at any concentration tested (Supplementary Fig. 9). Considering the subcellular location of Sam50 compared to BamA this is not entirely unexpected; Sam50 is separated from the extracellular space by the cytosol, cell membrane, and cell wall, whereas BamA in the outer membrane is easily accessible from the extracellular space. Imai et al. observed that treatment of mouse models with darobactin A was nontoxic and decreased the pathogen burden of mice infected with *E. coli*, *K. pneumoniae*, or *P. aeruginosa*[44]. This further supports that Sam50 in

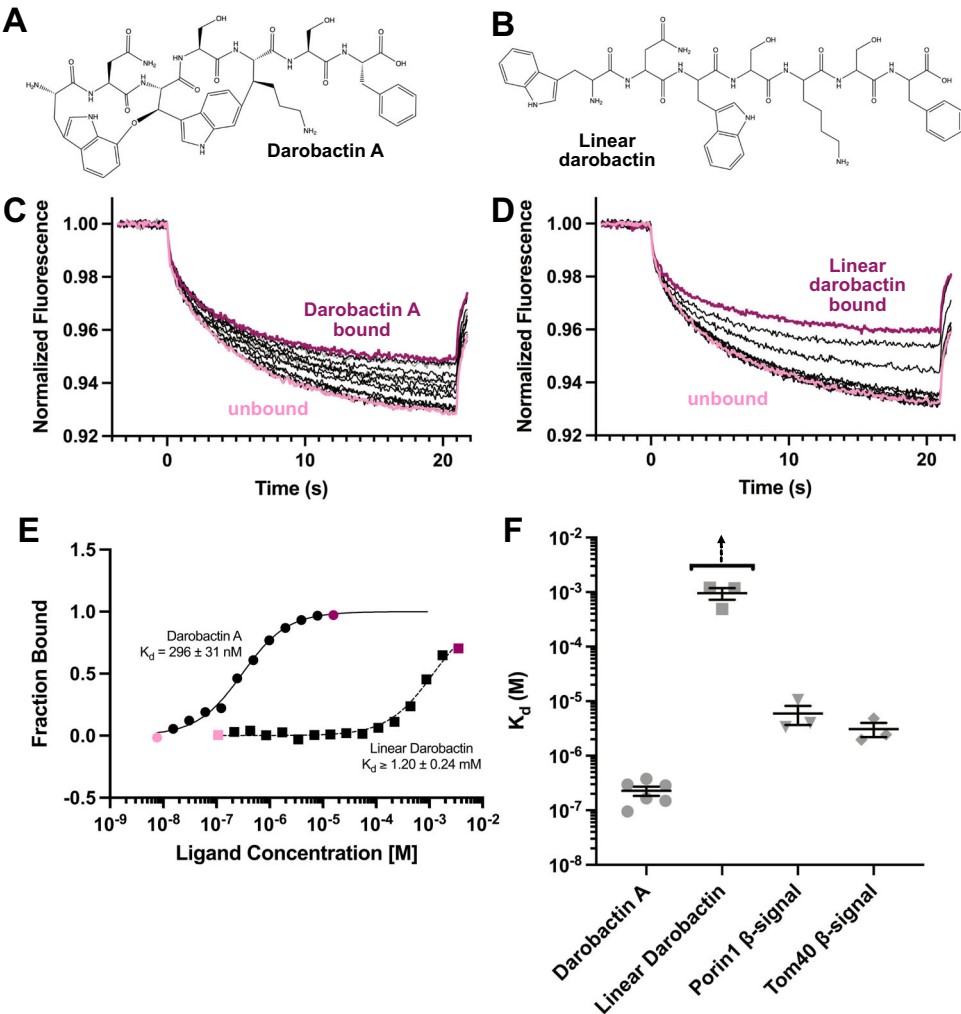

**Fig. 2 | Darobactin A binds the SAM complex with higher affinity than linear darobactin.** Chemical structures of **A** darobactin A and **B** linear darobactin. Representative MST fluorescence curves for one technical replicate of SAM complex titration with **C** darobactin A or **D** linear darobactin. Highest (dark pink) and lowest (light pink) tested ligand concentrations are indicated. Data are representative of three (linear darobactin) or six (darobactin A) independent replicates. **E** Merged dose response data from SAM complex binding darobactin A (circle data points, solid line fit) or linear darobactin (square data points, dashed line fit). Highest (dark pink) and lowest (light pink) tested ligand concentrations are indicated. Each plotted datapoint is an average of 2–3 technical replicates. Data are representative of three independent replicates. **F** Comparison of $K_d$ values

determined for $Tt$SAM complex interactions with darobactin A ($n = 6$), linear darobactin ($n = 3$), $Tt$Tom40 β-signal peptide ($n = 3$), and $Tt$Porin1 β-signal peptide ($n = 3$). Each point represents the mean $K_d$ of one biological replicate, calculated from 2 to 3 technical replicates. Error bars are calculated based on standard error of regression for each biological replicate. Highest concentrations of linear darobactin were not saturating, notated by arrow above linear darobactin data points. Data in (**C**–**E**) are representative for three (linear darobactin, Por1 β-signal peptide, and Tom40 β-signal peptide) or six (darobactin A) independent experiments. $K_d$ and fit values from all biological replicates can be found in Supplementary Table 1. Peptide information including sequences can be found in Supplementary Table 2. Replicate data and source data are provided as a Source Data file.

the mitochondrial membrane is not accessible to darobactin A supplied to the extracellular space.

### Initial structure reveals darobactin A binds to Sam50 β1

To better understand how darobactin A binds and inhibits SAM complex function in vitro from a structural standpoint, we used single particle cryo-EM to solve the structure of the *T. thermophilus* SAM complex bound to darobactin A. The SAM complex was purified as a monomer in LMNG/GDN, mixed with darobactin A in a 1:2 SAM:darobactin A molar ratio and incubated at 4 °C for approximately 41 hours before freezing cryo-EM grids.

We reconstructed an initial cryo-EM map of the SAM complex bound to darobactin A (SAM$^{daro}$) from 201,907 particles to a resolution of 2.89 Å (Supplementary Fig. 10). This resolution was sufficient to clearly assign density for Sam50, Sam35, Sam37, ergosterol molecules, as well as for darobactin A.

In our initial cryo-EM map, Sam50 β1-4 are rotated outward by approximately 45° to form an open lateral gate. There is additional density present proximal to Sam50 β1 that we attribute to darobactin A. However, due to the poor local resolution and the lack of density features in this region, we were not able to confidently build the darobactin A model, as several orientations fit within the density. Therefore, we generated models of each of the four darobactin A orientations with the terminal phenyl ring extended or condensed, for a total of eight models (E1-4, C1-4). We then refined these eight models against the density using Phenix and calculated the correlation coefficient (CC) for the darobactin A fit within the density (Supplementary Fig. 11A). Based on the CC score, where closer to 1 indicates a better fit of the model to the density, we determined that the C4 orientation was the best (Supplementary Fig. 11A).

We also employed Molecular Dynamics Flexible Fitting (MDFF) simulations to evaluate the different darobactin A binding

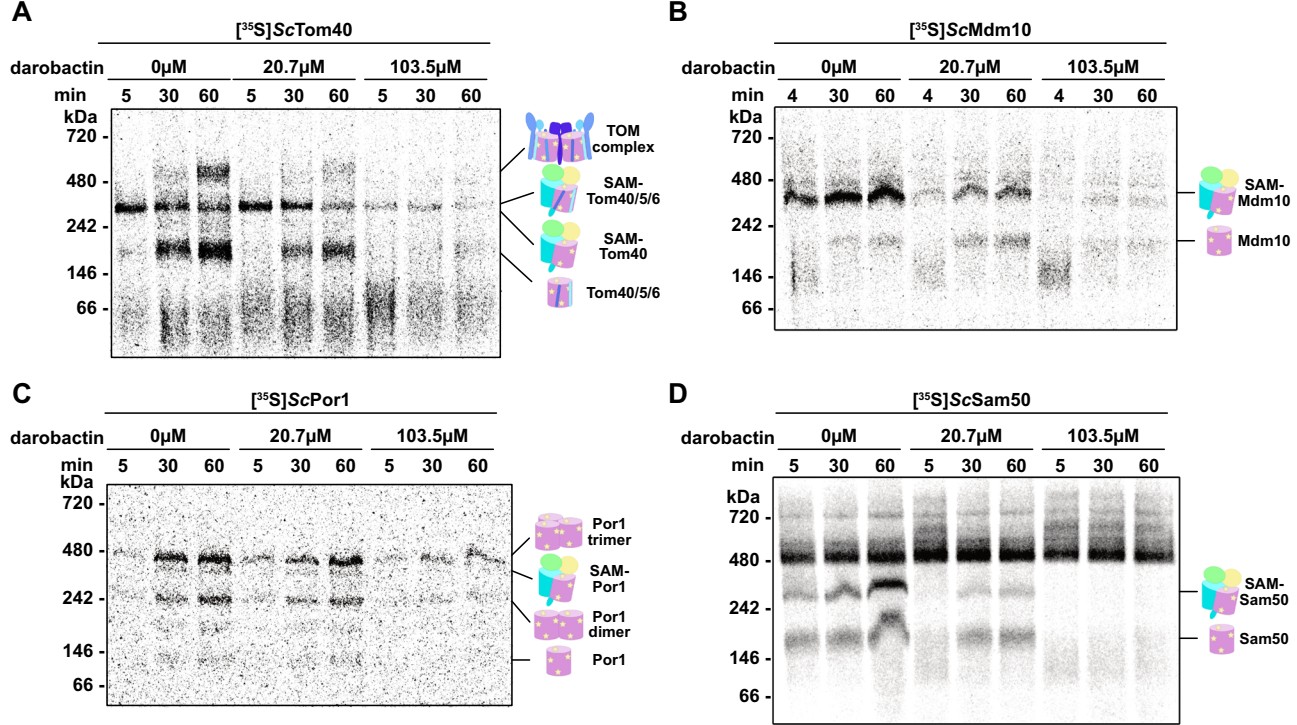

**Fig. 3 | Darobactin A inhibits mitochondrial β-barrel biogenesis in vitro.** Isolated wild-type *S. cerevisiae* mitochondria were incubated with increasing concentrations of darobactin A prior to incubation with translation lysate containing radiolabeled *S. cerevisiae* (**A**) Tom40, (**B**) Mdm10, (**C**) Porin1, or (**D**) Sam50. Following incubation, mitochondrial membranes were solubilized with digitonin, soluble fraction was subjected to BN-PAGE and autoradiography was used to identify protein complexes containing radiolabeled proteins. Data are representative of two (Sam50, Porin1) to three (Tom40, Mdm10) independent experiments. Replicate data and source data are provided as a Source Data file.

orientations. To address the missing regions in the cryo-EM structures, we used AlphaFold-predicted models to complete the protein structures before conducting the MDFF simulations. This method was applied to eight different initial refinements (E1, E2, E3, E4, C1, C2, C3, C4) to explore all possible orientations of darobactin A. We calculated the total interaction energy and the average number of hydrogen bonds between darobactin A and Sam50. The E4 and C4 conformations stand out in both metrics (Supplementary Fig. 11B,C). E4 has the highest average number of hydrogen bonds, likely because of its extended conformation, but C4 has the most favorable interaction energy overall, supporting it as the most likely orientation.

**Improved cryo-EM map confirms darobactin A orientation**

In an effort to obtain a cryo-EM map with improved density for darobactin A, we generated 2D templates from the initial map and reprocessed the cryo-EM data. One major population of the Sam50 lateral gate bound to darobactin A was separated using 3D classification (Supplementary Fig. 10). This final SAM^daro cryo-EM map was reconstructed from 163,391 particles to a resolution of 3.00 Å (Fig. 4, Table 1, Supplementary Figs. 10, 12). This map also contained clear density for Sam50, Sam35, Sam37, and darobactin A. The Sam37 transmembrane α-helix is visible in the SAM^daro data when the map is filtered to 10 Å resolution (Supplementary Fig. 12F) though this is lost at higher resolution, likely due to increased flexibility of this region.

Local resolution of the final SAM^daro map calculated in Phenix illustrates that the lateral gate region, including the density for darobactin A, still has lower resolution than the core of the SAM complex (Supplementary Fig. 12E). However, the darobactin A density contains more features than in the initial map, enough to model a single conformation. Encouragingly, the modeled darobactin A orientation from the final map is the same as the favored C4 binding mode previously

favored for the initial map. Given that these independent approaches all suggested the same binding mode of darobactin A, we are confident in our modeling of the darobactin A molecule in the final map. PyMOL analysis identified six hydrogen bonds stabilizing the interaction between darobactin A and Sam50 β1 (Fig. 4D). All six hydrogen bonds are between backbone residues on both molecules, which suggests that the exact amino acid sequence is not as critical as the overall secondary structure. The MST results presented earlier further support this, as the linear darobactin peptide exhibited a much weaker binding affinity than darobactin A despite both having the same amino acid sequence. Therefore, it is likely that the backbone cyclizations of darobactin A are restricting the secondary structure into a conformation that is favorable for stably interacting with the SAM complex.

**Comparison of the SAM complex with and without darobactin A**

In all three structures (SAM^op, SAM^cl, and SAM^daro), the organization of Sam35 and the cytosolic domain of Sam37 are quite similar, with C-alpha RMSDs below 0.87 Å (Supplementary Table 3). All three structures also contain ergosterol molecules in identical locations. In contrast, the Sam50 β-barrel is most similar between the SAM^daro and SAM^op structures, superimposing with C-alpha RMSD of 1.54 Å, while the C-alpha RMSD of the SAM^daro and SAM^cl structures is 2.79 Å (Supplementary Table 3).

The largest differences in the Sam50 subunit among these three structures are the conformations of the Sam50 β-barrel lateral gate and POTRA domain. The SAM^daro and SAM^op structures contain highly similar lateral gates with β1-β4 rotated outwards by approximately 45° and the POTRA domain very slightly shifted towards the Sam50 β-barrel lumen, compared to the SAM^cl structure (Fig. 5A–C, E). Since SAM^stall is the most open conformation, together with SAM^cl it can establish a convenient scale for the openness of the lateral gate. On this

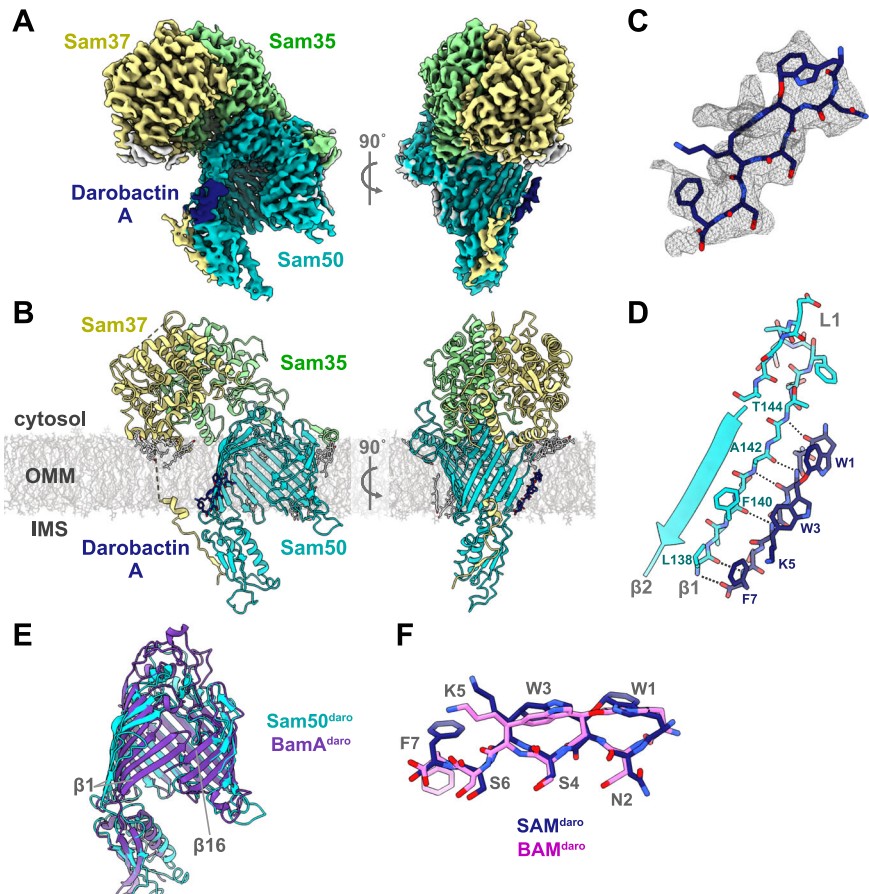

**Fig. 4 | Structure of the *T. thermophilus* SAM complex bound to darobactin A.**
**A** Cryo-EM density map of the *T. thermophilus* SAM complex (Sam50 in turquoise, Sam35 in green, Sam37 in yellow, lipid/detergent in gray) bound to darobactin A (dark blue). **B** Ribbon representation of the SAM complex bound to darobactin A in the context of a model lipid bilayer. **C** Darobactin A structure (dark blue) in the cryo-EM density map (gray mesh). **D** Close up of Sam50 β1 (turquoise) interactions with darobactin A (dark blue) viewed from the membrane plane (left) and interior of barrel (right). Interactions (gray dashed lines) identified by PyMOL (Version 2.4 Schrödinger, LLC) analysis. **E** Superposition of Sam50 from SAM[daro] structure (turquoise) and BamA from the BAM structure bound to darobactin A (purple, PDB: 7NRI) viewed from the membrane plane. Only BamA POTRA5 (POTRA closest to β-barrel) is displayed for clarity. **F** Superposition of Darobactin A from SAM[daro] (dark blue) and BAM[daro] (orchid) cryo-EM structures. Superpositions generated using ChimeraX v1.4 matchmaker[86,99]. See also Supplementary Figs. 10–12, 14.

scale, the lateral gate of Sam50 in our SAM[op] structure is 76–81% open (Fig. 5E, Table 2, Supplementary Fig. 13). This difference in Sam50 lateral gate positions could reflect different conformations upon β-signal binding or along the pathway from closed to open lateral gate. The remaining Sam50 β-strands (β5-16) and cytosolic loops are similar between all three structures (Fig. 5A–C, E). Additionally, the highly conserved Sam50 L6 adopts an almost identical conformation in all three structures (Fig. 5E), suggesting that it plays a role in stabilizing the Sam50 β-barrel.

### Comparison of Sam50 bound to darobactin A versus substrate

In all SAM complex structures published to date, the Sam50 lateral gate is in a semi-closed state, containing no hydrogen bonds between β1 and β16[17–19]. Several structures, including the structures of SAM[op] and SAM[daro] described here, contain Sam50 with β1-4 rotated outward away from β16 to open the β-barrel. The first of these cases is a non-physiological up-down dimer structure (two copies each Sam50, Sam35, and Sam37) produced during purification in which the β1 from each Sam50 were interacting[19]. The second instance of this opened Sam50 β-barrel is from the cryo-EM structure of the SAM complex stalled while folding Tom40 (SAM[stall])[34]. Of these structures, the SAM[stall] contains the widest open lateral gate (Fig. 5), while L1 and L2 of SAM[op] are 24.16% and 27.34% less open, and SAM[daro] L1 and L2 are 44.77% and 28.22% less open, respectively (Table 2, Supplementary Fig. 13). On the

scale of fully open SAM[stall] and fully closed SAM[cl], SAM[daro] lateral gate opening is most similar to SAM[op] (Fig. 5, Table 2, Supplementary Fig. 13). Despite these differences in the lateral gate conformation, the rest of the Sam50 β-barrel including highly conserved cytosolic loop 6 (L6) and POTRA domain are quite similar between the structures (Fig. 5E). The lack of conformational change in L6 despite drastic differences in lateral gate opening suggests that L6 plays a role in stabilizing the Sam50 β-barrel.

### Comparison of darobactin A bound Sam50 and BamA

Darobactin A interacts with β1 of both Sam50 and BamA (Fig. 4E, F, Supplementary Fig. 14). All structures of BamA bound to darobactin A contain darobactin A in the same orientation[47] and this binding orientation is consistent with β-signal interactions with Sam50 based on crosslinking results[9]. Additionally, the preferred binding mode of darobactin A based on our map is the same as that observed for BamA (Fig. 4F).

However, all structures published to date of the BAM complex bound to darobactin A contain a BamA β-barrel with a closed lateral gate stabilized by hydrogen bonds between β1 and 16 (Fig. 4E, Supplementary Fig. 14). In contrast, in our structure of the SAM complex bound to darobactin A, the Sam50 lateral gate is wide open with no interactions with β16 whatsoever (Fig. 4, Supplementary Fig. 14). Despite these differences in lateral gate conformations, the rest of the

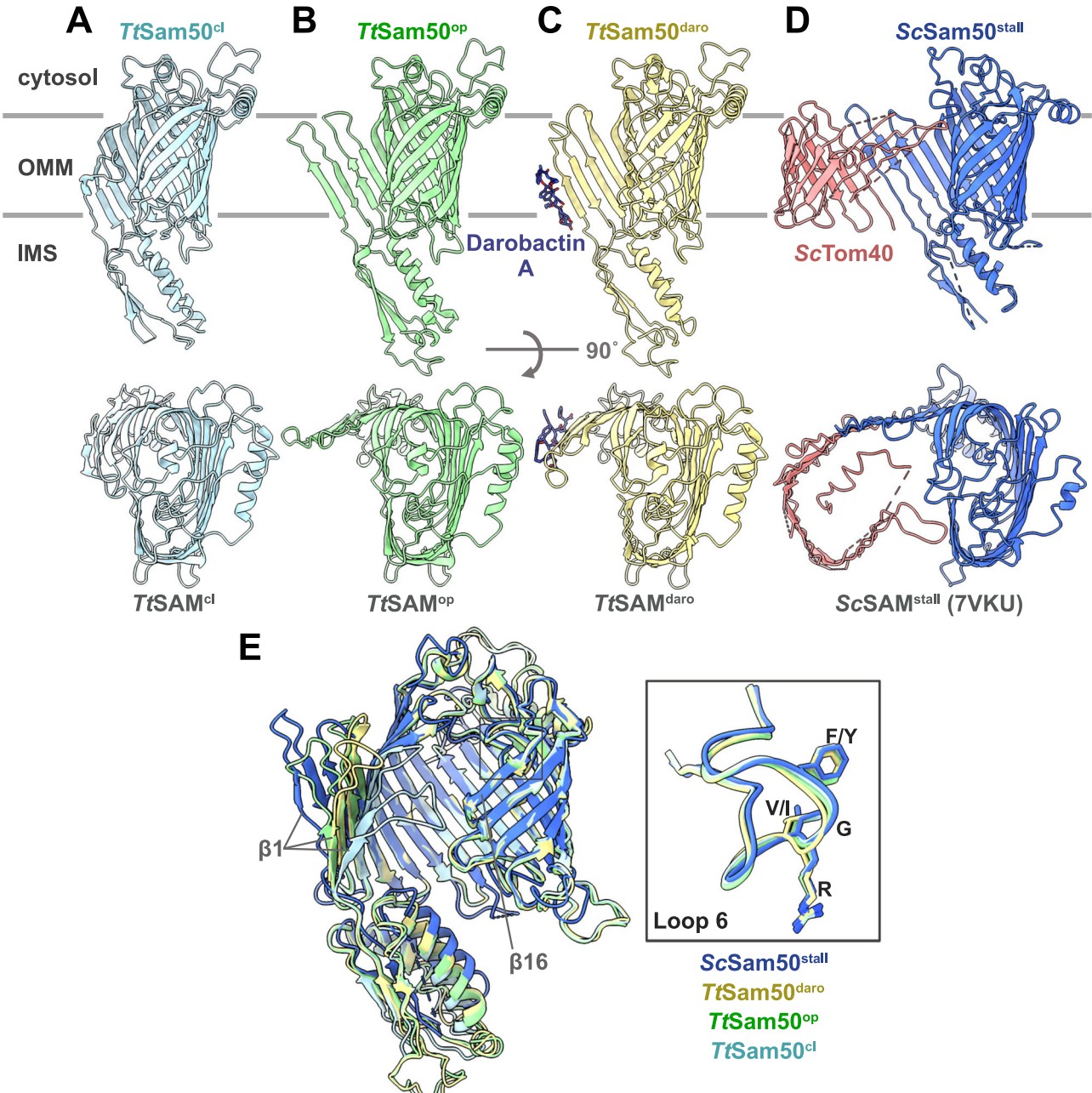

**Fig. 5 | Structural comparison of SAM complex interacting with darobactin A versus Tom40 folding intermediate. A** *Tt*Sam50[cl] structure (light blue) viewed from the side and top shows a closed, kidney bean-shaped β-barrel. **B** *Tt*Sam50[op] structure (light green) shows an open lateral gate with the first four β-strands rotated outwards. **C** *Tt*Sam50[daro] (Sam50 in khaki, darobactin A in dark blue) has outward rotation of the first four β-strands and darobactin A interacts with β1. **D** *Sc*Sam50[stall] (Sam50 in royal blue, Tom40 folding intermediate in salmon) has the first four β-strands of Sam50 rotate outwards to accommodate folding Tom40 protein (PDB: 7VKU). **E** Superposition of Sam50 from *Tt*SAM[cl] (light blue), *Tt*SAM[op] (light green), *Tt*SAM[daro] (khaki), and *Sc*SAM[stall] (royal blue, PDB: 7VKU) structures. Outward rotation of β1-4 observed in *Tt*SAM[op] (light green), *Tt*SAM[daro] (khaki), and *Sc*SAM[stall] (royal blue, PDB: 7VKU) structures, largest outward rotation in the *Sc*SAM[stall] structure. Gray box identifies cytosolic loop 6, inset panel shows close up with atom representation of (V/I)RG(F/Y) motif.

Sam50 and BamA β-barrel domains superimpose well (β-barrel domain C-alpha RMSD of 2.29 Å) (Fig. 4E). The Sam50 POTRA domain and BamA POTRA5 (the POTRA closest to the BamA β-barrel) are also in a very similar conformation and are not directly occluding the β-barrel lumen (Fig. 4E).

## Discussion

The SAM complex and BAM complex facilitate folding and insertion of β-barrel proteins into the outer membranes of mitochondria and Gram-negative bacteria, respectively. The core components of each

complex, Sam50 and BamA, share the greatest sequence and structural homology. Our structures of the SAM complex monomer in detergent suggest that the Sam50 lateral gate can dynamically sample open and closed conformations in the absence of substrate (Fig. 6A). Lateral gate opening in the absence of substrate has not been observed structurally for BamA. In fact, all structures of BamA published to date contain at least one hydrogen bond between β1 and β16 to stabilize a closed conformation of the β-barrel. The presence of these hydrogen bonds in the closed state would require energy to generate an open conformation. This would be favorable upon substrate binding, assuming

**Table 2 | Extent of gate loop opening in Sam50**

| Loop1 | SAM<sup>cl</sup> - SAM<sup>op</sup> | SAM<sup>cl</sup> - SAM<sup>daro</sup> | SAM<sup>op</sup> - SAM<sup>daro</sup> | Loop1 | SAM<sup>daro</sup> - SAM<sup>stall</sup> | SAM<sup>op</sup> - SAM<sup>stall</sup> | SAM<sup>cl</sup> - SAM<sup>stall</sup> |
|---|---|---|---|---|---|---|---|
| T144 - T144 | 14.25 | 10.93 | 5.15 | T144 - T129 | 8.58 | 4.30 | 18.45 |
| D145 - D145 | 19.25 | 12.99 | 7.97 | D145 - N130 | 10.78 | 3.22 | 22.27 |
| S152 - S152 | 12.13 | 11.34 | 3.28 | S152 - E138 | 7.55 | 6.09 | 18.14 |
| A153 - A153 | 10.93 | 9.60 | 3.39 | A153 - A139 | 6.20 | 4.26 | 15.09 |
| **average** | **14.14** | **11.21** | **4.94** | | **8.27** | **4.46** | **18.48** |
| Distance % | 76.48 | 60.66 | 26.76 | | 44.77 | 24.16 | 100.00 |
| Loop2 | SAM<sup>cl</sup> - SAM<sup>op</sup> | SAM<sup>cl</sup> - SAM<sup>daro</sup> | SAM<sup>op</sup> - SAM<sup>daro</sup> | Loop2 | SAM<sup>daro</sup> - SAM<sup>stall</sup> | SAM<sup>op</sup> - SAM<sup>stall</sup> | SAM<sup>cl</sup> - SAM<sup>stall</sup> |
| A173 - A173 | 4.64 | 4.74 | 1.15 | A173 - I162 | 1.63 | 1.69 | 5.99 |
| S174 - S174 | 6.30 | 5.98 | 1.11 | S174 - L163 | 2.64 | 2.10 | 8.36 |
| A182 - A182 | 4.56 | 4.76 | 0.30 | A182 - S171 | 1.31 | 1.29 | 4.93 |
| Y183 - Y183 | 2.91 | 3.11 | 0.44 | Y183 - F172 | 0.83 | 1.13 | 3.43 |
| **average** | **4.60** | **4.64** | **0.75** | | **1.60** | **1.55** | **5.67** |
| Distance % | 81.06 | 81.85 | 13.21 | | 28.22 | 27.34 | 100.00 |

Sam50 from each structure superimposed with SSM in Coot, distances (in Angstrom) between Calpha atoms of corresponding residues measured in Coot. Since the tips of the loops are more dynamic and have weak densities, residues with strong density located 4-5 residues inward from the tip were picked. Average distance for four residues shown in bold. % of distance calculated relative to the longest distance, between SAM<sup>cl</sup> and SAM<sup>stall</sup> (PDB ID:7VKU). See Supplementary Fig. 13 for the location of residues on each respective structure.

the substrate β-signal is binding with higher affinity than the affinity between the two sides of the lateral gate. In contrast, the closed conformation of Sam50 does not have any hydrogen bonds, therefore sampling different conformations would be more energetically favorable as no hydrogen bonds would be broken to open the gate.

A distinct reduction in membrane thickness at the Sam50 lateral gate compared to the average membrane thickness was observed over the course of the MD simulations presented here. Membrane thinning at the Sam50 and BamA lateral gates has been proposed to aid β-barrel insertion into the membrane by reducing the energetic barrier for membrane insertion[26,33,58–61]. In addition to aiding β-barrel insertion, this membrane destabilization may also aid in lateral gate opening with or without substrate.

The Sam50 lateral gate opening in the absence of substrate does bring into question what happens with the lumen of the barrel when the gate is open. It is likely that the hydrophilicity of the β-barrel lumen excludes lipid molecules from entering. In addition, the molecular dynamics simulations presented in this paper do not observe lipids in the β-barrel lumen, even with lateral gate opening. Considering the *S. cerevisiae* SAM structures contain a second β-barrel that faces the Sam50 lateral gate[17,18], it is possible that the second β-barrel could aid lateral gate opening by occluding lipid access to the Sam50 lumen. It should be noted however, that the SAM<sup>op</sup> structure presented here clashes with the second barrel in the *Sc*SAM structures. Therefore, it is likely that the *Sc*SAM complex could tolerate smaller lateral gate opening in the absence of substrate, but not as pronounced as the SAM<sup>op</sup> conformation observed here.

From all available structures[17,18] and AlphaFold models with a second β-barrel, it seems that the barrel is always mainly closed when a second full β-barrel is bound under Sam37 (Supplementary Fig. 15). When the barrel is fully open (SAM<sup>stall</sup>) then the second bound β-barrel is still unfolded/folding[34]. If it finished folding then it cannot be accommodated with the fully open Sam50 β-barrel, only with a more closed Sam50 β-barrel. The more open barrel is likely energetically favorable for binding substrate β-signals and starts folding them into a β-barrel. As the nascent β-barrel is growing the lateral gate will keep opening to accommodate the new barrel. This way the new barrel always shields the lateral gate, which should be energetically favorable. A fully open lateral gate with a nascent barrel with only a few beta strands would be energetically unfavorable, since the lateral gate would be too exposed to the lipid environment. The growing barrel may be slowly pushing the gate open, keeping the Sam50 lumen always shielded from the lipids.

The less dynamic lateral gate of BamA could be attributed to the difference in lipid environment compared to Sam50; the Gram-negative bacterial outer membrane is composed of an inner leaflet of phospholipids and an outer leaflet of lipopolysaccharides (LPS). The environment of the assembly machinery may also influence the Sam50 and BamA dynamics; Sam50 may sample more freely open and closed states with less severe consequences compared to BamA in the outer membrane, in which opening and allowing non-specific transport of molecules (including anti-microbial molecules) would be detrimental to the cell. These differences between SAM and BAM further suggest that while they have similar β-barrel cores and overall function, they likely have distinct functional mechanisms.

All SAM complex structures presented here contain densities that we have modeled as ergosterol, also confirmed by LC/MS analysis. Sterols are present in all eukaryotes, and play an important role in maintaining the fluidity, rigidity, and permeability of the membranes[62,63]. In vertebrates, cholesterol is the most abundant sterol, while ergosterol is the most abundant in lower eukaryotes such as yeast. While both mitochondrial membranes contain ergosterol, the ergosterol-to-phospholipid ratio is higher in the IMM than in the OMM, 7.0 wt% vs 2.1 wt%, respectively[49]. Despite the relatively low amounts of sterols in the OMM, another OMM protein, VDAC1 (the human homolog of Porin1), has been shown to have defined cholesterol binding sites[64,65] and alter the distribution of cholesterol in the mitochondrial membrane[66] as well as influence the local phospholipid distribution[67]. Therefore, we believe that a similar preferential localization of ergosterol to Sam50 is occurring in our samples, as all of the recombinant SAM complexes in our studies were expressed in and purified from yeast (*S. cerevisiae*) mitochondrial membranes.

We demonstrated that darobactin A binds to the SAM complex purified in detergent with nanomolar affinity. The linear form of darobactin A as well as linear β-signal peptides exhibited several orders of magnitude weaker interactions with the SAM complex. The differences in binding affinities between these peptides suggest that there is a secondary structure requirement for stable association of the peptide with the SAM complex. We also demonstrated that the presence of darobactin A inhibits the import and assembly of all four yeast mitochondrial β-barrel proteins in vitro. Our cryo-EM structure of the SAM complex bound to darobactin A suggests that this inhibition of SAM complex function is due to darobactin A stably associating with Sam50 β1 in the open conformation. The stable interaction between darobactin A and Sam50 β1 blocks substrate β-signal binding, therefore inhibiting β-barrel biogenesis (Fig. 6B).

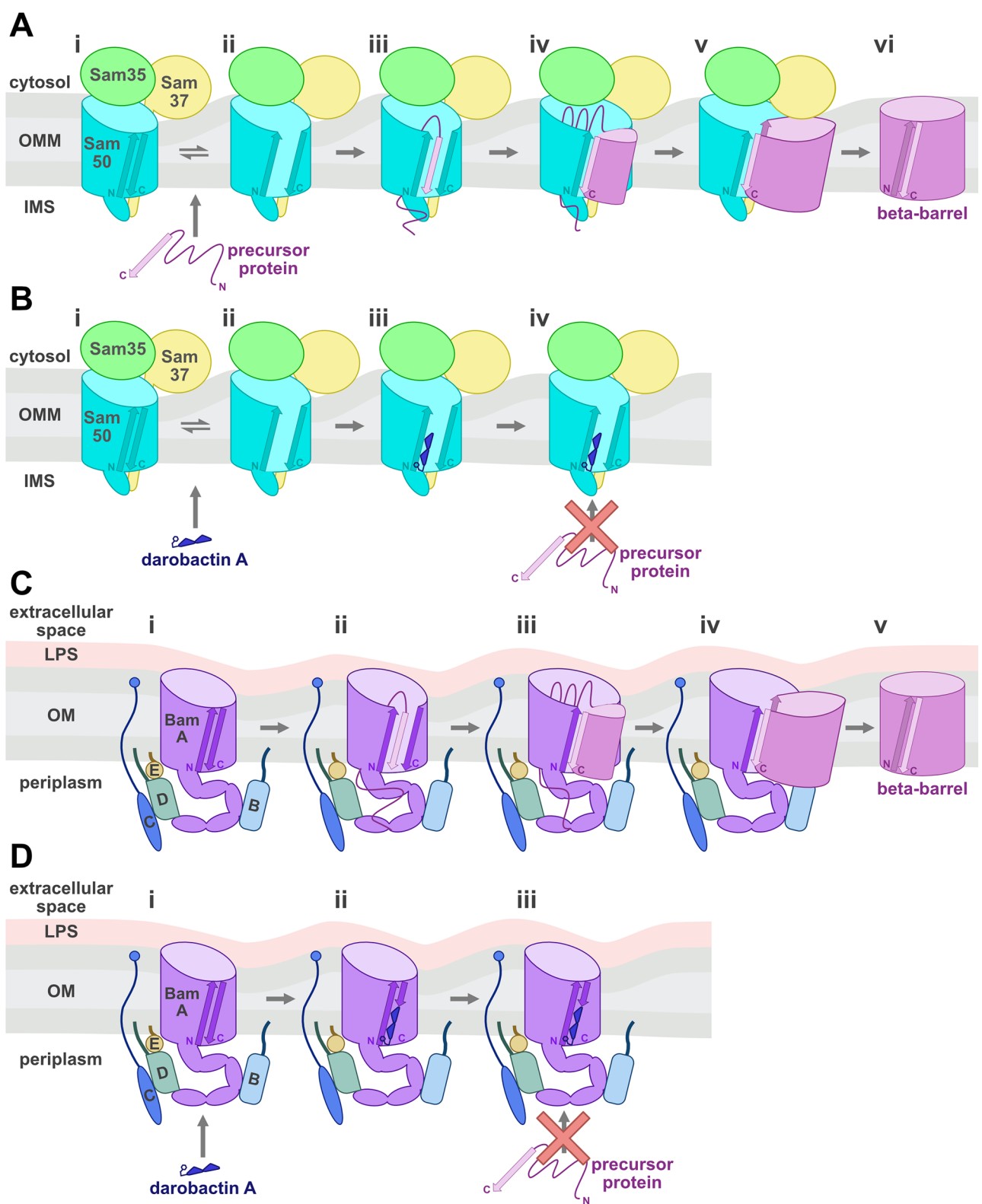

Our in vitro import assays demonstrate darobactin A dependent inhibition of the SAM complex in intact mitochondria isolated from *S. cerevisiae*. Seeing that the *S. cerevisiae* growth assays do not demonstrate sensitivity to darobactin A supplied in the growth media, we believe that darobactin A cannot penetrate the yeast cell wall. Cytotoxicity experiments demonstrated that darobactin A is nontoxic to several human cell lines as well as in mouse models[44]. This further supports that Sam50 in the mitochondrial membrane is not accessible

to darobactin A supplied to the extracellular space, of intact *S. cerevisiae*, human, or mouse cells. Therefore, darobactin A remains a viable candidate as an antibiotic to treat Gram-negative bacterial infections.

While the data presented here clearly support that darobactin A interacts similarly with Sam50 and BamA to inhibit β-barrel biogenesis, there is an important difference between the Sam50 and BamA lateral gate conformations. In the darobactin A bound structures, the BamA lateral gate remains partially closed while the Sam50 lateral gate is

**Fig. 6 | Proposed functional mechanism of darobactin A inhibition of SAM complex in vitro. A** In the absence of darobactin A, the Sam50 lateral gate dynamically opens and closes (i, ii). Mitochondrial outer membrane β-barrel precursor proteins containing a C-terminal β-signal are targeted to the SAM complex (i). The Sam50 lateral gate is stabilized in the open conformation by the β-signal binding to Sam50 β1 (iii). The precursor protein is hypothesized to fold by sequential β-hairpin insertions (iv). Sam37 stabilizes the precursor protein as it folds (v). After the final precursor protein β-strands are incorporated, the first pairs with the last, and the mature β-barrel is released laterally into the membrane (vi). **B** In contrast, the addition of darobactin A stabilizes the Sam50 lateral gate in the open conformation by binding to Sam50 β1 (iii). Darobactin A remains stably associated with Sam50 β1 which blocks the binding of precursor protein β-signal (iii) and therefore prevents assembly of the β-barrel protein into the outer membrane (iv). **C** BamA lateral gate remains closed in the absence of substrate (i). The BamA lateral gate opens to accommodate the β-signal binding to BamA β1 (ii). The precursor protein is hypothesized to fold by the β-hairpin insertions, and remains associated with the BAM complex until folding is complete (iii, iv). After the final precursor protein β-strand is incorporated, the mature β-barrel is released laterally into the membrane (v). **D** Darobactin A binds to BamA β1, however the BamA lateral gate remains closed through interactions between BamA β1 and β16 (i, ii). Darobactin A remains stably associated with BamA β1, blocking precursor β-signal binding and therefore inhibiting β-barrel biogenesis (iii). N = N-terminus, C = C-terminus.

much more open. This is consistent with the structures for each complex published to date, as SAM complex structures contain no hydrogen bonds to close the Sam50 lateral gate while BamA typically has at least one hydrogen bond closing the lateral gate[58,68]. Together these data suggest an evolutionary conserved mechanism for substrate β-signal recognition in mitochondria and Gram-negative bacteria, yet different dynamics of the lateral gate.

The function of the essential loop 6 in Sam50 and BamA remains unclear, as it has been shown to be required for Sam50 β-signal binding and insertion of subsequent β-hairpins[9,69] and proposed to stabilize the BamA β-barrel[26,33]. We observe no difference in loop 6 conformation between the SAM$^{cl}$, SAM$^{op}$, and SAM$^{daro}$ structures, nor with the SAM$^{stall}$ structure. Rather than playing an active role in β-signal binding or β-barrel folding, we propose that loop 6 instead is required to provide stability to the Sam50/BamA β-barrel while the first four β-strands open as the substrate folds.

The lack of conformational change in Sam35 and Sam37 subunits between the SAM$^{cl}$, SAM$^{op}$, and SAM$^{daro}$ structures does not lend any insight into their functional role in the complex. While Sam35 has been shown to specifically interact with the β-signal biochemically[12], we do not see evidence of Sam35 binding darobactin A in our SAM$^{daro}$ structure. Sam37 is proposed to aid in the later steps of β-barrel folding by stabilizing the newly forming β-barrel[12,16,17]. Since our SAM$^{daro}$ structure is at an early stage of the folding process and the fact darobactin A is only the size of a single β-strand, it is perhaps not surprising that Sam37 remains relatively static at this stage.

We present here structural and functional data to update the proposed functional model of the SAM complex in which in the absence of substrate, the Sam50 lateral gate can sample open and closed conformations. Further investigation of SAM complex structure and molecular mechanism throughout the entire folding process is required to fully understand the function of this essential protein complex. Of particular interest will be how the accessory proteins Sam35 and Sam37 contribute to the overall SAM complex function.

## Methods
### SAM complex expression plasmids
*Thermothelomyces thermophilus* Sam50, Sam37, and Sam35 coding sequences (Supplementary Table 4) were codon-optimized for yeast expression and cloned into pBEVY expression vectors after the GAL1 promoter. Affinity tags (10x His or TwinStrep), glycine/serine linkers, and TEV protease cleavage sites were included in some constructs. See Supplementary Table 5 for complete list of plasmids used in this study.

### Yeast transformation and growth
SAM complex plasmids were co-transformed (Supplementary Table 6) into *Saccharomyces cerevisiae* strain W303.1B (MATα {leu2-3,112 trp1-1 can1-100 ura3-1 ade2-1 his3-11,15}) using the lithium acetate/single-strand carrier DNA/PEG method[70–73]. Transformants were plated on selection agar (6.9 g/L yeast nitrogen base without amino acids, 0.62 g/L Clontech -Leu/-Trp/-Ura dropout supplement, 20 g/L bacto agar, 2% D-(+)-glucose) and incubated at 30 °C for 72 h. A small culture of selection media (6.9 g/L yeast nitrogen base without amino acids, 0.62 g/L Clontech -Leu/-Trp/-Ura dropout supplement, 2% D-(+)-glucose) was inoculated with one colony and grown overnight at 30 °C and 220 rpm. Overnight culture was used to inoculate 500 mL selection media and incubated again overnight at 30 °C and 220 rpm. YPG (10 g/L yeast extract, 20 g/L peptone, 3% glycerol, 0.1% glucose) cultures were inoculated to a start OD$_{600}$ around 0.18 and incubated at 30 °C and 220 rpm for 16 h. SAM complex expression was induced with D-galactose (0.4% final concentration) for 4 h. Cells were harvested, washed with cold ultra-pure water, and then the pellet was stored at −80 °C.

### Mitochondrial isolation for protein purification
Yeast cell pellet was thawed at 4 °C and resuspended in breaking buffer (650 mM sorbitol, 100 mM Tris-HCl, pH 8.0, 5 mM EDTA, pH 8.0, 5 mM amino hexanoic acid, 5 mM benzamidine, 0.2% BSA) and PMSF (2 mM final concentration) was added. Resuspended cells were passed through a Dyno-Mill Multi Lab (WAB) bead mill containing 0.5–0.75 µm glass beads at 35 mL/min with the chamber temperature maintained below 10 °C[74]. Lysed cells were collected on ice and 4 mL 200 mM PMSF added. Cell debris was removed from lysed cells with two low speed centrifugation steps (3470 × *g*, 30 min each, 4 °C), pouring supernatant into fresh centrifuge bottles after first spin. Mitochondrial membrane fraction was isolated by centrifugation (24,360 × *g*, 50 min, 4 °C), and resuspended in wash buffer (650 mM sorbitol, 100 mM Tris-HCl, pH 7.5, 5 mM amino hexanoic acid, 5 mM benzamidine) with a Dounce homogenizer. Mitochondrial membrane fraction was isolated by centrifugation (24,360 × *g*, 50 min, 4 °C), resuspended in Tris-buffered glycerol (TBG) buffer (100 mM Tris-HCl, pH 8.0, 10% glycerol) with Dounce homogenizer, and centrifuged again (24,360 × *g*, 50 min, 4 °C). Membrane pellet was resuspended and homogenized in TBG and separated into aliquots, which were snap frozen in liquid nitrogen prior to storage at −80 °C. A Pierce BCA Protein Assay Kit (Thermo Fischer Scientific) was used to determine the protein concentration of the mitochondrial membrane sample.

### SAM complex thermostability screen with MoltenProt
*T. thermophilus* SAM complex (Sam50 no tag, Sam35 no tag, Twin-Strep-GG-Sam37) was solubilized in 2% LMNG and purified by strep affinity chromatography and Superose 6 (Cytiva) size exclusion chromatography in 0.02% LMNG. Peak fraction from size exclusion chromatography (20 mM HEPES pH8, 150 mM NaCl, 0.02% LMNG) was kept at 4 °C and used for thermostability experiments. SAM complex sample was diluted and mixed with screen conditions from FEI88 (as described in ref. 75), RUBIC Additive (provided by SPC facility at EMBL Hamburg[76]), and a custom detergent screen (Loew lab, EMBL Hamburg). All protein samples were centrifuged (16,000 × *g*, 10 min, 4 °C) to remove potential aggregates, and loaded in standard glass capillaries. Measurements were collected with Prometheus NT.48 (NanoTemper Technologies) at SPC facility in EMBL Hamburg, with the laser emission pre-adjusted to get fluorescence readings above 2000 RFU. Samples were measured in temperature range 20–95 °C with

temperature slope of 1 °C/min. All screens included a triplicate of the sample in the original buffer and a water control, where an equal volume of ultra-pure water was added instead of the screen. Curve fitting and computation of thermodynamic parameters were performed with MoltenProt[48,77]. Addition of 500 mM arginine increased the aggregation temperature of the sample. It was previously demonstrated that addition of arginine or glutamate enhances solubility of proteins during purification without negative effects on functionality[78]. 200 mM arginine was chosen for purification of monomer SAM.

## Monomer SAM complex protein purification
One mitochondrial membrane aliquot was thawed and adjusted to 10 mg/mL with TBG. Sodium chloride (150 mM final concentration), arginine (200 mM final concentration), and two Roche cOmplete Protease Inhibitor Cocktail tablets were added. The sample was stirred at 4 °C until protease tablets dissolved. For MST and SAM-darobactin A cryo-EM experiments, the sample was solubilized by adding 2% (final concentration) LMNG (Anatrace) dropwise and stirring for 1.5 h at 4 °C. For SAM complex monomer cryo-EM experiments in the absence of Darobactin A and lipid LC/MS, sample was solubilized by adding 2% (final concentration) GDN (Anatrace) dropwise and stirring for 17.5 h stirring at 4 °C. Following membrane solubilization, soluble material was isolated by ultracentrifugation (208,000 × g, 45 min, 4 °C). The supernatant was filtered with a 0.22 μm SteriFlip vacuum filter (Millipore) before adding equilibrated Strep-Tactin Sepharose (IBA GmbH) resin and rocking at 4 °C for 3–4 h. Supernatant and strep resin were then transferred to gravity column and flow through was collected. Resin was washed with at least 6 column volumes of wash buffer (100 mM Tris-HCl, pH 8.0, 150 mM NaCl, 1 mM EDTA, pH 8.0, 200 mM arginine, 0.02% GDN). SAM complex was eluted with elution buffer (100 mM Tris-HCl, pH 8.0, 150 mM NaCl, 1 mM EDTA, pH 8.0, 2.5 mM Desthiobiotin, 200 mM arginine, 0.02% GDN). Fractions containing SAM complex were concentrated with 100 kDa molecular weight cut-off Amicon Ultra Centrifugal Filter Unit (Millipore). Concentrated sample was injected onto Superose 6 10/300 column (Cytiva) at 0.12 mL/min in size-exclusion buffer (20 mM HEPES pH 8.0, 150 mM NaCl, 0.02% GDN). SDS-PAGE and BN-PAGE were used to evaluate protein sample.

## Darobactin A isolation
Darobactin A was isolated as described in Imai et al. 2019. Briefly, the *P. khanii* strain was grown for 8–10 days in tryptic soy broth. Culture supernatant was obtained by centrifugation and subjected to capture by polymer resin XAD16N. The compound was eluted from XAD16N by 50% methanol with 0.1% (v/v) formic acid and concentrated using a rotary evaporator. The concentrated elution was subjected to strong cation exchange chromatography (SP Sepharose XL, GE Healthcare). The pH7 fraction containing darobactin A was concentrated, followed by isolation using RP-HPLC on semi-preparative C18 column by water and acetonitrile with 0.1% formic acid (v/v) gradient (Agilent 1260 HPLC, C18 5 μm; 250 mm × 10 mm). Darobactin A chemical structure figure (Fig. 2A) was made with ChemDraw 21.0.0.

## Peptides
Linear darobactin and β-signal peptides were purchased from GenScript. Sequence information for linear peptides can be found in Supplementary Table 2. Linear darobactin chemical structure figure (Fig. 2B) was made with ChemDraw 21.0.0.

## SAM monomer cryo-EM data collection
SAM complex from the size-exclusion chromatography peak fraction was concentrated to ~6.4 mg/mL. A 3 μL aliquot of SAM monomer sample was applied to freshly glow-discharged holey carbon grids (Quantifoil R1.2/1.3, copper, 300 mesh), incubated for 0 s, blotted for

5 s (blot force 5) then plunge-frozen in liquid ethane with a FEI Vitrobot Mark IV plunger (22 °C, 100% humidity).

Cryo-EM data were collected at the NIH Multi Institute Cryo-EM Facility (MICEF) on a Titian Krios G4 electron microscope (Thermo-Fisher) operated at 300 kV with a Gatan Bioquantum-K3 with 20 eV energy slit. Micrographs were collected at 105,000× nominal magnification, with 0.412 Å/super-res pix, 40 frames, $70e^-/\text{Å}^2$ total dose, defocus range of −0.6 to −1.6 μm. Parameters also listed in Table 1.

## SAM monomer cryo-EM image processing
All frames were motion corrected and binned by 2 using cryoSPARC patch motion correction, and patch CTF estimation was completed in cryoSPARC v4.4.0[79]. Micrographs were manually curated to exclude micrographs based on CTF fit resolution (worse than 4.5 Å), relative ice thickness (greater than 1.15), and total full-frame motion distance (greater than 50pix). The resulting 13,205 accepted micrographs were used for further processing.

Initial $SAM^{op}$ and $SAM^{cl}$ 3D references were generated using standard single-particle data analysis in cryoSPARC. The initial 3D references were then used to generate templates for template particle picking. Multiple iterations of 2D classification were completed, followed by ab-initio reconstruction and heterogeneous refinement. 2D classifications were used to rescue good particles from sub-optimal heterogeneous refinement classes. Best classes from heterogeneous refinement and selected 2D classes were pooled and subjected to further rounds of ab-initio reconstruction, heterogeneous refinement, and 2D classification. The final best class was used in non-uniform refinement[80].

A mask of the Sam50 lateral gate and POTRA domain was generated from 6WUT and 6WUM pdbs using molmap in Chimera v1.15[81], and volume tools in cryoSPARC. The non-uniform refinement particles and solvent mask, along with the lateral gate focus mask were input into a 3D classification job with force hard classification parameter on. Ab-initio reconstructions were generated for the two largest classes, followed by non-uniform refinements. Cryo-EM data processing workflow is summarized in Supplementary Fig. 5.

## Darobactin A bound SAM complex cryo-EM data collection
SAM complex from size-exclusion chromatography peak fraction was concentrated to ~4.76 mg/mL. Concentrated SAM complex was incubated with darobactin A in a 1 SAM: 2 darobactin A molar ratio for approximately 41 h at 4 °C. A 3 μL aliquot of SAM-darobactin A sample was applied to freshly glow-discharged holey carbon grids (Quantifoil R1.2/1.3, copper, 300 mesh), incubated for 5 s, blotted for 5 s (blot force 5), then plunge-frozen in liquid ethane with a FEI Vitrobot Mark IV plunger (22 °C, 100% humidity).

Cryo-EM data were collected at the NIH Intramural Cryo-EM Facility (NICE) on a Titian Krios G3 electron microscope (Thermo-Fisher) operated at 300 kV with a Gatan Bioquantum-K3 with 20 eV energy slit. Micrographs were collected at 105,000× with 0.415 Å/super-res pix, 50 frames, $54.4e^-/\text{Å}^2$ total dose, defocus range of −0.7 to −2.0 μm. Parameters also listed in Table 1.

## Darobactin A bound SAM complex cryo-EM image processing
All frames were motion corrected and binned by 2 using cryoSPARC patch motion correction, and patch CTF estimation was completed in cryoSPARC v4.4.1[79]. Micrographs were manually curated to exclude micrographs based on CTF fit resolution (worse than 4.5 Å), relative ice thickness (greater than 1.15), total full-frame motion distance (greater than 217pix), and total full-frame motion curvature (greater than 50). The resulting 9235 accepted micrographs were used for further processing.

An initial model was generated using standard single-particle data analysis. Templates generated from initial 3D reference were used for template particle picking. Multiple iterations of 2D classification were

completed, followed by ab-initio reconstruction and heterogeneous refinement. Best classes from heterogeneous refinement were pooled and subjected to further rounds of ab-initio reconstruction and heterogeneous refinement. The final best class was used in non-uniform refinement[80]. A mask of Darobactin A with the Sam50 lateral gate and POTRA domain was generated from an initial model using molmap in Chimera v1.15[81], and volume tools in cryoSPARC. The nonuniform refinement particles and solvent mask, as well as the lateral gate focus mask, were input into a 3D classification job with force hard classification parameter on. Particles from the largest class were then used in an ab-initio reconstruction, followed by nonuniform refinements. Cryo-EM data processing workflow is summarized in Supplementary Fig. 10.

## Model building, refinement, and analysis

Models were built in Coot[82] and refined with Phenix v1.21[83]. The final model was refined using Isolde v1.8[84] followed by another Phenix refinement. The interaction analyses for darobactin A and Sam50 were completed using QT PISA[85] and PyMOL v2.4.1 (Schrödinger, LLC). Local resolution was calculated in Phenix v1.20[83]. Structure figures were made using Chimera v1.15 and ChimeraX v1.7[81,86]. Superposition figures were created using the Matchmaker tool in ChimeraX v1.7[86]. RMSD values were calculated using Superpose from CCP4i v8. Overall RMSD values calculated by Superpose using secondary structure matching, while specific chain RMSD values by specified atoms/residues mode.

## Microscale thermophoresis (MST) binding assays

Purified SAM complex was diluted to 200 nM with size exclusion buffer, mixed with 100 nM Monolith RED-tris-NTA 2nd Generation (NanoTemper Technologies) and incubated for 30 min at room temperature to label. Following labeling incubation, sample was centrifuged (15,000 × g, 10 min, 4 °C) to remove aggregates. MST binding affinity titration samples were made with a serial dilution of peptide and constant labeled SAM complex (final concentration 25 nM). Samples were loaded into Monolith premium capillaries (NanoTemper Technologies) and MST measured using a Monolith NT.115 (NanoTemper Technologies) with fluorescence intensity settings determined from pretest experiment. Data were analyzed using MO. Affinity Analysis software (NanoTemper Technologies), and final plots were generated in GraphPad Prism 9.

## Yeast growth and mitochondrial isolation for import assay

50 mL YPG media (10 g/L yeast extract, 20 g/L peptone, 3% glycerol) in a 125 mL flask was inoculated with one wild type W303.1B colony (Supplementary Table 6) and incubated overnight at 30 °C and 220 rpm. The overnight culture was used to inoculate 1.5 L YPG (10 g/L yeast extract, 20 g/L peptone, 3% glycerol) to a start $OD_{600}$ of ~0.02 and incubated at 30 °C and 220 rpm until an $OD_{600}$ of 1 was reached (~19.5 h). Cells were harvested by centrifugation (5500 × g, 20 °C, 8 min) and washed with Milli-Q water, then centrifuged (2500 × g, 20 °C, 8 min).

Mitochondria were isolated following protocol from Priesnitz, Pfanner & Becker[53]. Briefly, cell pellet was resuspended in pre-warmed DTT buffer (0.1 M Tris/$H_2SO_4$ pH9.4, 10 mM DTT) (2 mL buffer per 1 g cells) and incubated for 30 min at 30 °C and 220 rpm. Cells were isolated by centrifugation (2500 × g, 20 °C, 5 min) and supernatant discarded. Zymolyase buffer (20 mM KPi pH7.4, 1.2 M sorbitol) containing Zymolyase 20 T (4 mg per 1 g cells) was used to resuspend cells (7 mL buffer per 1 mL cells), then incubated at 30 °C and 220 rpm for 35 min. Spheroplasts were isolated by centrifugation (2500 × g, 20 °C, 5 min), then washed in Zymolyase buffer and isolated by centrifugation (2500 × g, 20 °C, 5 min). Supernatant was discarded and pellet was resuspended in chilled homogenization buffer containing fresh PMSF (0.6 M sorbitol, 10 mM Tris/HCl pH7.4, 1 mM ETDA, 1 mM PMSF, 0.2% BSA) (6.5 mL per 1 g cells). Resuspension was homogenized by

eighteen strokes of a Dounce homogenizer on ice. Cell debris was removed by centrifugation (2500 × g, 4 °C, 5 min). Mitochondria were isolated by centrifugation (17,000 × g, 4 °C, 15 min). Mitochondrial pellet was resuspended in SEM buffer (250 mM sucrose, 1 mM EDTA, 10 mM MOPS/KOH pH7.2) then centrifuged to remove remaining cell debris (2500 × g, 4 °C, 5 min). Mitochondria were re-isolated by centrifugation (17,000 × g, 4 °C, 15 min). Mitochondrial pellet was resuspended in a small volume of SEM buffer, and total protein concentration was determined using Pierce BCA Protein Assay Kit (Thermo Fischer Scientific). Concentration was adjusted to be ~10 mg/mL with SEM buffer, aliquots of 50–100 μL were made and flash frozen in liquid nitrogen before storing at −80 °C.

## Synthesis of radiolabeled substrates

*Sc*Tom40, *Sc*Mdm10, *Sc*Por1 and *Sc*Sam50 coding sequences (Supplementary Table 4) were cloned into pGEM-4Z (Promega) following the SP6 promoter (Supplementary Table 5) by GenScript or LifeSct LLC. DNA was linearized by PCR (Supplementary Table 7) and capped mRNA generated with mMESSAGE mMACHINE SP6 transcription kit (Life Technologies) and purified using the MEGAclear kit (Life Technologies). Purified mRNA was added to Flexi Rabbit Reticulocyte Lysate System (Promega) following enclosed instructions with the addition of 80 mM potassium chloride which aided in increased expression of target substrates. Lysate reactions were incubated at 30 °C, 300 rpm for 90 min. Alternatively, circular DNA template was used in the TNT Coupled Reticulocyte Lysate SP6 System (Promega) with 0.1–0.4 mM magnesium acetate, incubated at 25 °C or 30 °C, 300 rpm for 90 min. Following synthesis with either system, reactions were placed on ice and 5 mM final cold methionine and 250 mM final sucrose were added to the lysate. Lysate fractions were snap frozen in liquid nitrogen and stored at −20 °C.

## Mitochondrial import assay

Import assay protocol was adapted from ref. 53. Mitochondrial import reactions were prepared in 1.5 mL Eppendorf tubes to a final volume of 50 μL and kept on ice. Each import reaction contained 3% (v/v) essentially fatty acid free BSA, 250 mM sucrose, 80 mM KCl, 5 mM $MgCl_2$, 2 mM $KH_2PO_4$, 5 mM methionine, 10 mM MOPS/KOH pH 7.2, 4 mM NADH, 4 mM ATP, 5 mM Creatine phosphate, 0.1 mg/mL creatine kinase, 1% ethanol, and 1 mg/mL mitochondria. Darobactin A was added to a final concentration of 20.7 μM (20 μg/mL) or 103.5 μM (100 μg/mL), or the same volume of water added for the control samples. After the addition of darobactin A or water, import reactions were incubated on ice for 2 min, then incubated at 25 °C for 2 min before adding 8–10% (v/v) lysate containing [$S^{35}$]-Met-labeled precursor protein. The longest time point samples were started first. Import was terminated by placing the samples on ice, adding AVO mix (final concentration of 8 μM Antimycin A from *Streptomyces* sp., 1 μM valinomycin, and 20 μM Oligomycin from *Streptomyces diastatochromogenes*). Samples were then centrifuged (13,000 × g, 10 min, 4 °C), supernatant removed, and mitochondrial pellet resuspended in 50 μL SEM buffer (250 mM sucrose, 1 mM EDTA, 10 mM MOPS/KOH pH7.2). Mitochondria were again isolated by centrifugation (13,000 × g, 10 min, 4 °C) and the supernatant removed. The mitochondrial pellet was resuspended in solubilization buffer (1.5% (w/v) digitonin, 50 mM NaCl, 20 mM Tris/HCl pH 7.4, 0.1 mM EDTA, 10% (v/v) glycerol, 1 mM PMSF) and incubated on ice for 15 min. Following solubilization, samples were centrifuged (13,000 × g, 10 min, 4 °C), and the soluble supernatant saved. BN-PAGE samples (soluble fraction, 1x loading buffer, 0.5% G-250 additive) were prepared and a NativePAGE gel (Invitrogen) prepared with dark blue cathode buffer and 1x running buffer. BN-PAGE was run at 150 V for 1 h, then 250 V for 30–60 min. Gel was fixed (40% methanol, 10% acetic acid), destained (8% acetic acid), and rinsed with water before drying to Whatman filter paper using BioRad Model 583 gel dryer. Dried gel was wrapped in Saran wrap,

taped to an exposure cassette, and exposed to Fuji Imaging Plate BAS-IP MS 2040E for 20–24 h at room temperature. Radiolabeled proteins were detected using a Fujifilm FLA-9000 Phosphorimager.

### *S. cerevisiae* growth assay

W303.1B glycerol stock was streaked onto YPD agar (10 g/L yeast extract, 20 g/L peptone, 20 g/L bacto agar, 2% D-(+)-glucose) and incubated at 30 °C for 3 days. Overnight YPD (10 g/L yeast extract, 20 g/L peptone, 2% D-(+)-glucose) and YPG (10 g/L yeast extract, 20 g/L peptone, 3% glycerol) cultures were inoculated with 1 colony each and incubated at 30 °C for ~16 h. Fresh media was inoculated overnight to a start $OD_{600}$ of 0.3 for YPD or 0.6 for YPG and allowed to grow to an $OD_{600}$ of ~1. Culture was then diluted to an $OD_{600}$ of 0.1 in fresh media. Darobactin A was sterile filtered, serially diluted, and pipetted into a sterile 96 well plate with low evaporation lid (Corning Costar). *S. cerevisiae* culture was added to each well for a final volume of 200 µL. Final concentrations of Darobactin A ranged from 0–128 µg/mL. A ClarioStar Plus plate reader was used to measure the $OD_{600}$ of each well every 30 min with temperature control at 30 °C on and 200 rpm double orbital shaking between reads. Data were collected for 24 h and plotted with GraphPad Prism 9.

### Molecular dynamics simulations

The Molecular Dynamics (MD) simulations of the Sam50 β-barrel and SAM complex were performed in a realistic mitochondrial outer membrane environment. Starting from the high-resolution structures in PDB IDs 6WUT, 6WUH, 7BTX and 7E4H, either the complete SAM complex was built or just the Sam50 β-barrel. All missing residues in the protein structures were built with either AlphaFold2 (AF2) or SWISS-MODEL[87,88]. The outer mitochondrial membrane model is symmetric and contains 16% LLPC, 5% YOPC, 8% PLPC, 8% POPC, 10% DOPC, 11% LLPE, 3% YOPE, 7% PLPE, 6% POPE, 7% DOPE, 10% DLiPl, 4% cardiolipin, 4% SLPA, and 1% PLPS[89–92]. In total, we built six systems and simulated two replicas of each. These systems included two of the SAM complexes from *T. thermophilus* (PDB IDs: 6WUH and 6WUT) and four Sam50 β-barrel-only simulations from *T. thermophilus* (PDB IDs: 6WUT and 6WUH) and *S. cerevisiae* (PDB IDs: 7BTX and 7E4H). We performed 40 ns of equilibration, gradually releasing all restraints, followed by 5-µs production runs for each replica using the CHARMM36m force field and NAMD, resulting in a total simulation time of 60 µs. A 4-fs integration timestep was used during production runs, enabled by hydrogen mass repartitioning. A Langevin thermostat and barostat were employed with a damping coefficient of 1 ps$^{-1}$. Non-bonded interactions were treated with a cutoff of 12 Å. To monitor membrane thickness, we calculated the distance between the head groups of lipids using the LOOS package[93] for each point in a 45 × 45 grid on the membrane plane of each system (see Source Data file). For membrane thickness around the lateral gate, we computed the average thickness over a 20 × 20 Å$^2$ region near the lateral gate and compared it to the average thickness across the entire 90 × 90 Å$^2$ membrane (Supplementary Fig. 8).

The initial structure of the SAM complex bound to darobactin A was obtained from cryo-EM data; however, the orientation of darobactin A was uncertain. To determine the most likely orientation, we used Molecular Dynamics Flexible Fitting (MDFF)[94]. For each of the eight complexes with different darobactin A orientations refined from cryo-EM data, missing regions were modeled using AF2 predictions to generate a complete initial model. The cryo-EM density map was used to apply additional forces (Gscale 0.3) to the atoms whose positions were determined from the cryo-EM data, guiding them into and holding them in the density map. No additional force was applied to the first loop of Sam50 due to the low electron density in the map for that region. All MDFF simulations were performed in a vacuum with a dielectric constant of 80 to partially mimic a solvated environment using the CHARMM36m force field for proteins[95] with the darobactin A

parameters obtained from a previous study[96], and executed using the NAMD software package[97]. Secondary structure restraints were applied to preserve helices and β-sheets. Energy minimization was followed by MDFF for 2.5 ns. All systems were built, and simulations were analyzed using Visual Molecular Dynamics (VMD)[98]. For hydrogen bond calculations, we set the distance cut-off at 3.5 Å and the angle cut-off of 30°. Interaction energies between darobactin and Sam50 were computed using VMD's namdenergy plugin to evaluate non-bonded interactions.

### Ergosterol LC/MS analysis

Liquid chromatography mass spectrometry (LC-MS) grade water and methanol were purchased through Fisher Scientific (Waltham, MA) and an ergosterol standard was purchased through Cayman Chemical (Ann Arbor, MI). To precipitate proteins and isolate ergosterol, 600 µL of cold methanol was added to 200 µL of each protein aliquot and stored at −80 °C overnight. Samples were centrifuged at 16,000 × *g* for 10 min at 4 °C. The supernatant was transferred to an LC-MS vial prior to LC-MS analysis. All standard dilutions were prepared similarly samples.

All LCMS analysis utilized a LD40 X3 UHPLC (Shimadzu Co.) and a 6500 + QTRAP (AB Sciex Pte. Ltd.) mass spectrometer. A method for ergosterol quantification was developed via fragmentation of a pure standard in positive ionization mode. Four multiple-reaction monitoring (MRM) parent-daughter ion pairs were tested for signal-to-noise in a mixed biological matrix and 379.6->171.4, with a collision cell voltage of 25 V, was used for subsequent quantification. To facilitate chromatographic separation, samples were injected onto a Waters™ XBridge BEH C8 column (3.0 × 50 mm) and eluted with a gradient from water with 0.01% acetic acid to methanol with 0.01% acetic acid over 13 min. Samples were analyzed as single injections. Peaks were integrated and concentration was established using a non-weighted linear fit to a half-log spaced standard curve using SciexOS 3.1 (AB Sciex Pte. Ltd.).

### Reporting summary

Further information on research design is available in the Nature Portfolio Reporting Summary linked to this article.

## Data availability

Atomic coordinates for the cryo-EM structures described here have been deposited in the Protein Data Bank under accession codes 9NK6 (SAM$^{cl}$), 9NK7 (SAM$^{op}$), and 9NK8 (SAM$^{daro}$). The cryo-EM 3D maps have been deposited in Electron Microscopy Data Bank under accession codes EMD-49494 (SAM$^{cl}$), EMD-49495 (SAM$^{op}$), and EMD-49496 (SAM$^{daro}$). The MD simulation input, output, and parameter files have been deposited on Zenodo (https://doi.org/10.5281/zenodo.17479556). PDB codes of previously published structures used in this study are 7VKU, 6WUH, 6WUT, 7BTX, 7E4H, 6WUM and 7NRI. The full gel images, raw intensities for ergosterol MS measurements, and replicate data are provided in the Source Data file. Additional source data for all figures and files is available from the authors upon request. Source data are provided with this paper.

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

## Acknowledgements

This work utilized the NHLBI Biophysics Core Facility, NIH Intramural Cryo-EM Facility (NICE), NIH Multi-Institute Cryo-EM Facility (MICEF), the computational resources of the NIH HPC Biowulf cluster (http://hpc.nih.gov), and was supported by the NIH-Metabolomics Consortia. We thank Rick Huang, AJ Morton, and Yanxiang Cui for help with cryo-EM data collection on the Titan Krios Electron Microscopes. We thank Grzegorz Piszczek and Di Wu for technical training and support for biophysical data collection. We thank Harris Bernstein and Janine Peterson for their advice on the radiolabeled import assay experiments and for allowing us to use their gel drying and imaging equipment. We thank the Jinwei Zhang lab for the use of their ClarioStar Plus plate reader. We thank the Fred Dyda lab for the use of computational resources. We thank Anna Ratliff for providing the negative control protein sample for ergosterol LC/MS analysis. This research was supported by the Intramural Research Program of the National Institute of Diabetes and Digestive and Kidney Diseases (K.A.D., I.B., S.H., H.C., and S.K.B.; NIDDK), the National Institute of Neurological Disorders and Stroke (J.A.M.; NINDS), and the Division of Intramural Research of the National Institute of Allergy and Infectious Disease (G.C. and B.S.; NIAID) within the National Institutes of Health (NIH). The contributions of the NIH author(s) are considered Works of the United States Government. The findings and conclusions presented in this paper are those of the author(s) and do not necessarily reflect the views of the NIH or the U.S. Department of Health and Human Services. J.C.G. acknowledges NIH for support (R01-GM148586). K.L. and A.I. acknowledge the support from NIH grant R01AI158388. Computational resources used for MD simulations were provided through ACCESS (grant TG-MCB130173), which is supported by NSF grants 2138259, 2138286, 2138307, 2137603, and 2138296. V.K. was supported by Boehringer Ingelheim Fonds PhD fellowship. We acknowledge technical support by the SPC facility at EMBL Hamburg and thank Christian Loew for providing the detergent screen for thermostability assays. This project was supported by funds available to T.C.M. at the Institute of Microbial and Molecular Sciences of the University Medical Center Hamburg-Eppendorf (UKE).

## Author contributions

K.A.D., J.A.M. and S.K.B. designed the study. V.K. and T.C.M. conducted SAM complex thermostability screens. K.A.D. and S.H. purified samples, froze cryo-EM grids, and conducted MST experiments. K.A.D. and H.C. collected and processed cryo-EM data. K.A.D. conducted mitochondrial import assays and yeast growth assays. I.B. built, refined, and analyzed the structures. K.L. and A.I. isolated darobactin A G.G., K.K. and J.C.G. conducted and analyzed MDFF and MD simulations. G.C. and B.S. conducted LC/MS experiments and analysis. K.A.D. wrote the first draft of the manuscript. All authors analyzed the data and edited the manuscript.

## Competing interests

The authors declare no competing interests.
