## [Transparent Peer Review file · Nature Communications]

The dynamic lateral gate of the mitochondrial β -barrel biogenesis machinery is blocked by darobactin A

Corresponding Author: Dr Susan Buchanan

Version 0:

Reviewer comments:

Reviewer #1

(Remarks to the Author)

In this manuscript Diederichs et al. present single particle cryo-EM structures of the SAM complex with Sam50 in open, closed, and darobactin-bound states. Using a biochemically optimized sample, the authors show co-existence of SAM complexes with the lateral gate of Sam50 open and closed, which suggests that the lateral gate is intrinsically dynamic. The authors also performed simulations to investigate lateral gate dynamics and observed membrane thinning at the gate, consistent with previous observations in bacterial BamA. The authors then show, using binding assays and cryo-EM, that Sam50 β 1 strand binds to darobactin, an antibiotic peptide that has been shown to inhibit outer membrane beta-barrel insertion by BamA in Gram-negative bacteria. In the SAM-darobactin structure, the Sam50 lateral gate is stabilised in the open conformation, in contrast to BamA-darobactin where the gate is closed. In vitro data show that darobactin inhibits SAM, though in vivo there seems to be no effect.

Collectively, the novel Sam50 lateral gate dynamics presented in this manuscript provide significant mechanistic insights into the function of the SAM complex, and how it differs from the related BAM complex in Gram-negative bacteria.

Main concerns/suggestions:

1. The quality of the density is not good enough to justify ergosterol placement in any of the cryo-EM maps. The authors use GDN in their sample preparations, so those densities could correspond to the steroidal moiety of GDN. Also, some of the ergosterol molecules are modelled with their hydroxyl groups pointing towards the hydrophobic interior of the micelle/membrane. Can the authors provide any orthogonal information that would support the lipid density assignment, e.g. native mass spectrometry or lipid analysis of their cryo-EM samples? Was ergosterol included in the molecular dynamics simulation setup, and did it occupy any of the putative binding sites observed in the cryo-EM maps? In the absence of additional evidence, I strongly suggest omitting the ergosterol molecules from the models.
2. The authors suggest that Porin1 might use a "slightly different" folding mechanism to other MOM proteins based on the results of the in vitro folding assay (L238-241). Are there any sequence features, e.g. differences in beta-signal composition, that would support this?
3. The SAM complex was incubated with darobactin A for an extended time prior to freezing grids (L260). Why was such a long incubation time required, given the relatively high affinity of the complex for darobactin A as determined by MST? Did the authors initially attempt a shorter incubation time but didn't see any bound darobactin in their cryo-EM data?
4. The darobactin density in the cryo-EM maps is poor, and the authors use a sensible strategy to propose the most likely binding mode. I strongly suggest including a panel showing the map-model fit for the darobactin and Sam50 in the main figures (e.g. Figure 4C) for full disclosure that the density is not great.
5. It is not clear if the region where darobactin binds is mobile or if darobactin binds in multiple similar conformations, resulting in poor density. I suggest running MD simulations with the darobactin-bound Sam50 embedded in a membrane to look at the dynamics of the lateral gate and darobactin.
6. Have the authors tested the levels of SAM substrates in *S. cerevisiae* grown in the presence of darobactin

(Supplementary Figure 8)? It is possible that indeed SAM function is impaired but not enough to result in a growth defect.

7. L392-398: how do the authors envisage initiation of nascent substrate insertion into the membrane by SAM if the lateral gate is blocked by another barrel?

8. L399: can the authors please show some of these AlphaFold models?

9. Figure 6: I suggest that the authors include a schematic of BamA-mediated β -barrel insertion for comparison and highlight the differences in the mechanism of nascent substrate insertion initiation to better convey the main finding of the paper.

10. There should be a paragraph in the Discussion about the in vivo relevance of the authors' findings. Darobactin inhibits SAM in vitro but not in vivo, at least in *S. cerevisiae*. Could darobactin be active in other eukaryotes, or is this purely an in vitro effect?

Minor comments:

Figure 1F: what was the distance cut-off for the hydrogen bonds?

Figure 1G: it would be useful to have a structural comparison of this partially closed lateral gate conformation with the fully closed Sam50 structure.

Figure 5E: I recommend changing the colour scheme in this figure and using different colours rather than shades of blue/green. It's quite difficult to tell the structures apart when superposed.

Table 2: it would be helpful for the reader to show on a structure where these residues are located.

Supplementary Figure 11: the units for membrane thickness are missing.

Supplementary Figure 12: can the authors please show a superposition of the two structures? It's difficult to compare them as presently shown.

Reviewer #2

(Remarks to the Author)

The SAM complex imports mitochondrial beta-barrel proteins into the outer membrane. The beta-barrel protein Sam50 is the core subunit and releases proteins laterally into the target membrane. The lateral gate is formed between the first and last beta-strand of Sam50. The mechanism of lateral opening of the Sam50 beta-barrel is poorly understood. The manuscript by Diederichs and colleagues provide interesting insights into these steps by structural investigations of the SAM complex. They report two structures of the SAM complex, a closed form similar to the published structure and a partially open form of the Sam50 beta-barrel within the SAM complex. They used Darobactin a for functional studies of the lateral gate, which is a cyclic peptide that contains a beta-signal. They show that Darobactin a binds to the SAM complex and inhibits import of beta-barrel proteins into mitochondria. A cryoEM structure revealed that the peptide binds to the first beta-strand at the lateral gate and thereby stabilizes the open state of the Sam50 beta-barrel.

This study provides exciting new insights into the function of Sam50 in the lateral release of beta-barrel proteins, which are important contributions to the protein sorting field. The presented experiments and structures are of high quality and well described. After minor revision the manuscript would be a good candidate for Nat. Commun..

Specific comments

In Figure 3, the authors convincingly show that the import of beta-barrel proteins is inhibited upon Darobactin a treatment. Is this effect specific for beta-barrel proteins or is the import of other mitochondrial proteins (e.g. presequence or carrier pathway) also affected?

Does Darobactin a compete with the binding of the beta-barrel signal to the SAM complex? The authors could investigate whether Darobactin a inhibits binding of beta-barrel precursor proteins to the SAM complex. One possibility would be to import Tom40G354A variant that arrest at the SAM complex.

The authors identified ergosterol in the structure of the SAM complex, which is surprising since ergosterol is not a major component of mitochondrial membranes. In this context, I recommend to extend the discussion of phospholipids and their role for the SAM complex.

Reviewer #3

(Remarks to the Author)

The manuscript "The dynamic lateral gate of the mitochondrial β -barrel biogenesis machinery is blocked by darobactin A" by authors Diederichs et al. is a well-executed study on the structure and function of the mitochondrial outer membrane protein

assembly machinery. The results add important new information to our understanding of the SAM complex and its inhibition, and the manuscript is very well written.

It is not completely clear to me what the authors aimed for by their molecular dynamics simulations, as most beta-barrels in their simulations seem to show a transition state between the open and closed barrels while not providing further information on the functional mechanism of SAM. Also, I find the h-bond counting a bit confusing, for example in the caption to Figure 1G "Sam50 in a partially closed conformation, observed with, at most, a couple of hydrogen bonds". Does that mean hydrogen bonds are formed during the transition from the closed to the open state? This is in contrast with statements made in the discussion, e.g. "the closed conformation of Sam50 does not have any hydrogen bonds, therefore sampling different conformations would be more energetically favorable as no hydrogen bonds would be broken to open the gate". Please can the authors clarify?

Irrespective of the fact that more could have been achieved in the MD simulations in my view, the observations that lipids cannot enter at least the semi-open state of the barrel and that the gate region leads to a strong thinning of the surrounding membrane are interesting and important. In the MD methods, information about the force field used and some other important MD parameters seems to be missing, however. I also wondered, as the authors mentioned that simulations were carried out in a realistic mitochondrial membrane model, what the actual composition of the membrane was. They only provide a literature citation of the measured composition - is their model of the exact same composition?

Overall though, the manuscript is well-rounded and provides a wealth of interesting and new information.

Version 1:

Reviewer comments:

Reviewer #1

(Remarks to the Author)

Thanks to the authors for thoughtfully addressing my and other reviewer's comments. The new LC-MS lipid analysis convincingly shows the presence of ergosterol in the purified SAM complex, and I have no further issues with including the ergosterol in the models. Changes to the figures and the text are appropriate and improve clarity. I recommend publication of the revised manuscript.

Reviewer #2

(Remarks to the Author)

The revised version of the manuscript by Diederichs and colleagues improved. However, it is not entirely clear whether incubation of mitochondria with Darobactin specifically affects the import of beta-barrel proteins and not the import of other mitochondrial proteins like presequence-containing precursor protein. Except of this point, the authors addressed my concerns. The findings are interesting and the manuscript is suitable for a broad readership of Nature Communication.

Reviewer #3

(Remarks to the Author)

The authors have made adequate changes to their manuscript. I recommend publication.

Nature Communications Manuscript Submission NCOMMS-25-17724

Authors' Responses to Reviewer Comments

Reviewer #1 (Remarks to the Author):

In this manuscript Diederichs et al. present single particle cryo-EM structures of the SAM complex with Sam50 in open, closed, and darobactin-bound states. Using a biochemically optimized sample, the authors show co-existence of SAM complexes with the lateral gate of Sam50 open and closed, which suggests that the lateral gate is intrinsically dynamic. The authors also performed simulations to investigate lateral gate dynamics and observed membrane thinning at the gate, consistent with previous observations in bacterial BamA. The authors then show, using binding assays and cryo-EM, that Sam50 β 1 strand binds to darobactin, an antibiotic peptide that has been shown to inhibit outer membrane beta-barrel insertion by BamA in Gram-negative bacteria. In the SAM-darobactin structure, the Sam50 lateral gate is stabilised in the open conformation, in contrast to BamA-darobactin where the gate is closed. In vitro data show that darobactin inhibits SAM, though in vivo there seems to be no effect.

Collectively, the novel Sam50 lateral gate dynamics presented in this manuscript provide significant mechanistic insights into the function of the SAM complex, and how it differs from the related BAM complex in Gram-negative bacteria.

Main concerns/suggestions:

1. The quality of the density is not good enough to justify ergosterol placement in any of the cryo-EM maps. The authors use GDN in their sample preparations, so those densities could correspond to the steroidal moiety of GDN.

We first tried placing GDN as the ligand in these densities, however the majority of the ligand densities show only a sterol moiety and for the ones with extending density from the sterol the glycosyl moieties of GDN could not be plausibly fit. Assuming that only the diosgenin part of GDN is visible, the 12 ergosterol molecules replaced with diosgenin refine to a CC of 0.62.

The average CC of the 12 ergosterol ligands is 0.68 in the 3Å SAM-darobactin A map. The CC of darobactin is 0.78 and the overall CC is 0.87. While the density is uneven for the 12 ergosterol ligands this is not exactly a “poor” fit into the map. In fact, the CC of the ergosterol ligands (0.68) is much better than that of the diosgenin parts of GDN (0.62) further supporting our decision to model ergosterol in these densities.

Also, some of the ergosterol molecules are modelled with their hydroxyl groups pointing towards the hydrophobic interior of the micelle/membrane.

Different orientations were tried and refined for the ligands, and the best density fit (CC) was chosen for each. The micelle is not a native membrane environment, so we are not sure all ligands would be “properly” oriented. There are clear ligand densities that are

not perpendicular to the membrane plane, with a perpendicular orientation to other ligands.

Can the authors provide any orthogonal information that would support the lipid density assignment, e.g. native mass spectrometry or lipid analysis of their cryo-EM samples?

It is true that native yeast mitochondrial membranes contain very low amounts of ergosterol. All recombinant SAM complexes were expressed and purified from yeast, ergosterol being possibly captured and enriched in the sample.

We have conducted lipid LC/MS analysis on the SAM complex purified in GDN following the same purification protocol used for the preparation of our cryo-EM samples. This analysis confirmed that indeed ergosterol is present in our purified SAM complex protein sample (see Supplementary Figure 7). A bacterial OM β -barrel protein OprM purified in GDN using the same facility and columns did not contain ergosterol, further indicating that the ergosterol observed in the SAM complex is specific.

It is known that another mitochondrial outer membrane protein, VDAC1 (the human homolog of Porin1), has defined cholesterol binding sites (Budelier *et al.*, 2017, main text ref. 64; Cheng *et al.*, 2019, ref. 65) and alters the distribution of cholesterol in the mitochondrial membrane (Campbell & Chan, 2007, ref. 66). Recent MD simulations have revealed that VDAC influences the local lipid distribution as well; with cholesterol and phosphatidylethanolamine (PE) preferentially surrounding and contacting the VDAC β -barrel, while the second lipid layer (layer adjacent to the PE and cholesterol layer) primarily contained phosphatidylcholine (PC) (Lafargue *et al.*, 2024, ref. 67). Taken together, these observations of cholesterol preferentially interacting with VDAC despite the low cholesterol concentration in the mitochondrial outer membrane further support our placement of ergosterol (the sterol of yeast mitochondrial membranes) in our cryoEM densities, as a similar preferential localization of ergosterol to Sam50 could be occurring.

Was ergosterol included in the molecular dynamics simulation setup, and did it occupy any of the putative binding sites observed in the cryo-EM maps?

Ergosterol was not the focus of this investigation and was not included in the MD simulations. It does not occupy any putative binding sites.

In the absence of additional evidence, I strongly suggest omitting the ergosterol molecules from the models.

Since the sterol densities are clear in all our maps and the LC/MS data clearly demonstrate ergosterol is present in our purified protein sample, we have kept the ergosterol molecules in our models.

2. The authors suggest that Porin1 might use a “slightly different” folding mechanism to other MOM proteins based on the results of the in vitro folding assay (L238-241). Are there any sequence features, e.g. differences in beta-signal composition, that would support this?

There are no obvious differences in the β -signal sequences of Porin1/VDAC compared to other mitochondrial β -barrels (see Figure 2H of Kutik *et al.*, 2008 (ref. 12), which compares β -signal sequences from all four mitochondrial β -barrel proteins).

In a 2007 JMB paper, Jörg Kleinschmidt's group demonstrated that urea unfolded VDAC1 (the human Porin1 homolog) spontaneously folded into lipid bilayers in the absence of the SAM complex (Shanmugavadivu *et al.*, 2007, ref. 57). This, in addition to our results of the *in vitro* folding assay, further supports the possibility of alternative folding mechanisms for Porin1.

To clarify this, we have edited the text to: " Given the high abundance of Porin1 relative to other mitochondrial outer membrane β -barrels (Morgenstern *et al.*, 2017), the observation of human VDAC1 (the human Porin1 homolog) to spontaneously fold into lipid bilayers *in vitro* (Shanmugavadivu *et al.*, 2007), and the different impact of darobactin A on Porin1 assembly, it is possible that the folding mechanism of Porin1 is slightly different from other β -barrels, and/or Porin1 might use a few different folding mechanisms."

3. The SAM complex was incubated with darobactin A for an extended time prior to freezing grids (L260). Why was such a long incubation time required, given the relatively high affinity of the complex for darobactin A as determined by MST? Did the authors initially attempt a shorter incubation time but didn't see any bound darobactin in their cryo-EM data?

The long incubation time prior to freezing grids was not an experimental requirement, rather a result of schedule conflicts between end of purification and instrument availability to freeze grids. We did not attempt a shorter incubation time before freezing grids.

4. The darobactin density in the cryo-EM maps is poor, and the authors use a sensible strategy to propose the most likely binding mode. I strongly suggest including a panel showing the map-model fit for the darobactin and Sam50 in the main figures (e.g. Figure 4C) for full disclosure that the density is not great.

Thank you for this suggestion, we have updated Figure 4 to include this map-model fit.

5. It is not clear if the region where darobactin binds is mobile or if darobactin binds in multiple similar conformations, resulting in poor density. I suggest running MD simulations with the darobactin-bound Sam50 embedded in a membrane to look at the dynamics of the lateral gate and darobactin.

As suggested, we performed additional molecular dynamics simulations of the darobactin-bound SAM complex embedded in a membrane. We used the same membrane composition and CHARMM36 force field as in our previous simulations. Two independent replicas of 1 μ s each were run. The simulations show that the lateral gate is still mobile, transferring from approximately the SAM^{dar} to the SAM^{cl} conformation. However, darobactin remains bound in the same position throughout the simulations.

The interaction energy between darobactin and Sam50 also indicates that the binding is stable. In the figure, the black dotted line represents the average interaction energy from the C4 conformation, while the bars for rep1 and rep2 show average interaction energy over the last 0.5 μ s of simulation, demonstrating that the interaction energy improves over the course of the simulations (see Supplementary Fig. 10D).

6. Have the authors tested the levels of SAM substrates in *S. cerevisiae* grown in the presence of darobactin (Supplementary Figure 8)? It is possible that indeed SAM function is impaired but not enough to result in a growth defect.

We have not tested the levels of SAM substrates in *S. cerevisiae* grown in the presence of darobactin. Given that Sam50 is essential for cell viability and β -barrel biogenesis, and has low endogenous expression levels (1,500 copies of Sam50 per yeast cell) compared to other outer membrane β -barrel proteins (Tom40 has 19,500 and VDAC has 630,000 copies per cell) (Morgenstern *et al.*, 2017, ref. 56; Pfanner *et al.*, 2019, ref. 5), we expect that even minor impairment of the SAM complex function *in vivo* would result in an observable growth defect. Our *in vitro* import assays demonstrate darobactin A dependent inhibition of the SAM complex in intact mitochondria isolated from *S. cerevisiae*. Seeing that the *S. cerevisiae* growth assays do not demonstrate sensitivity to darobactin A supplied in the growth media, we believe that darobactin A cannot permeate the yeast cell wall.

7. L392-398: how do the authors envisage initiation of nascent substrate insertion into the membrane by SAM if the lateral gate is blocked by another barrel?

Takeda *et al.* propose a β -barrel switching mechanism for SAM complex function based on their *S. cerevisiae* SAM complex structures containing two β -barrels (Takeda *et al.*, 2021, ref. 17). In this proposed mechanism, the precursor protein binds Sam50a in a complex of SAM monomer (one copy each of Sam50, Sam35, and Sam37) + an additional copy of Sam50 ("Sam50b"). The precursor protein would be sequentially folded through a series of β -hairpin insertion events (the budding mechanism). At some point, the folding β -barrel will reach a size that displaces Sam50b from the complex and the folding β -barrel assumes the position under Sam37 which provides additional stability during folding. After the β -barrel is fully folded, it can be released in several different ways depending on the protein. High abundance proteins, such as Porin1, could spontaneously release into the membrane. Lower abundance and/or slower folding proteins, such as Tom40, may require an additional protein, Mdm10, to physically displace it from the SAM complex monomer. Mdm10 bound SAM complex could then be regenerated to the initial SAM+Sam50b complex through another copy of Sam50 coming in and displacing Mdm10.

8. L399: can the authors please show some of these AlphaFold models?

We have added Supplementary Figure 14 to show these.

9. Figure 6: I suggest that the authors include a schematic of BamA-mediated β -barrel insertion for comparison and highlight the differences in the mechanism of nascent substrate insertion initiation to better convey the main finding of the paper.

We have added two panels to Figure 6 showing BAM complex mechanism in the absence (6C) and presence (6D) of darobactin A for comparison with the previously included SAM complex schematics. While several mechanisms have been proposed for BAM complex function, only one is shown here for simplicity and to maintain the focus of the manuscript on the SAM complex.

10. There should be a paragraph in the Discussion about the in vivo relevance of the authors' findings. Darobactin inhibits SAM in vitro but not in vivo, at least in *S. cerevisiae*. Could darobactin be active in other eukaryotes, or is this purely an in vitro effect?

Imai *et al.*, 2019 (ref. 44) tested cytotoxicity of darobactin A on several human cell lines, and found it was not toxic. Additionally, they observed that treatment of mouse models with darobactin A was nontoxic and decreased the pathogen burden of mice infected with *E. coli*, *K. pneumoniae*, or *P. aeruginosa*. This further supports that Sam50 in the mitochondrial membrane is not accessible to darobactin A supplied to the extracellular space, of intact *S. cerevisiae*, human, or mouse cells. We have added a paragraph in the discussion on this topic, as requested.

Minor comments:

Figure 1F: what was the distance cut-off for the hydrogen bonds?

3.5 Å. We added a few sentences at the end of the molecular dynamics methods section (now page 38).

Figure 1G: it would be useful to have a structural comparison of this partially closed lateral gate conformation with the fully closed Sam50 structure.

We have updated the caption of 1G be called "zipped closed" instead of "partially closed".

Figure 5E: I recommend changing the colour scheme in this figure and using different colours rather than shades of blue/green. It's quite difficult to tell the structures apart when superposed.

Thank you for this suggestion. We have re-made the figure with a new color scheme to make discerning the structures easier.

Table 2: it would be helpful for the reader to show on a structure where these residues are located.

Thank you for the suggestion. We have added a new figure, Supplementary Figure 13, to highlight these residues. We also added the statement "See Supplementary Figure 13 for location of residues on each respective structure." to our Table 2 caption to direct readers to this added figure.

Supplementary Figure 11: the units for membrane thickness are missing.

We believe you are referring to Supplementary Figure 7 (now Supplementary Figure 8), the plot of average membrane thickness measurements from MD simulations. If so, this has been fixed.

Supplementary Figure 12: can the authors please show a superposition of the two structures? It's difficult to compare them as presently shown.

Figure 4F contains the superposition that you request. The purpose of this supplementary figure is to show the two structures separately. We have added "See also Figure 4E for superposition of these structures." to the legend (now Supplementary Figure 14) to direct readers to the corresponding superposition.

Reviewer #2 (Remarks to the Author):

The SAM complex imports mitochondrial beta-barrel proteins into the outer membrane. The beta-barrel protein Sam50 is the core subunit and releases proteins laterally into the target membrane. The lateral gate is formed between the first and last beta-strand of Sam50. The mechanism of lateral opening of the Sam50 beta-barrel is poorly understood. The manuscript by Diederichs and colleagues provide interesting insights into these steps by structural investigations of the SAM complex. They report two structures of the SAM complex, a closed form similar to the published structure and a partially open form of the Sam50 beta-barrel within the SAM complex. They used Darobactin a for functional studies of the lateral gate, which is a cyclic peptide that contains a beta-signal. They show that Darobactin a binds to the SAM complex and inhibits import of beta-barrel proteins into mitochondria. A cryoEM structure revealed that the peptide binds to the first beta-strand at the lateral gate and thereby stabilizes the open state of the Sam50 beta-barrel.

This study provides exciting new insights into the function of Sam50 in the lateral release of beta-barrel proteins, which are important contributions to the protein sorting field. The presented experiments and structures are of high quality and well described. After minor revision the manuscript would be a good candidate for Nat. Commun.

Specific comments

In Figure 3, the authors convincingly show that the import of beta-barrel proteins is inhibited upon Darobactin a treatment. Is this effect specific for beta-barrel proteins or is

the import of other mitochondrial proteins (e.g. presequence or carrier pathway) also affected?

We would like to clarify that the SAM complex assembles and inserts β -barrel proteins into the mitochondrial outer membrane (Supplementary Figure 1A, step iii). A different protein complex, the translocase of the outer membrane (TOM) complex, imports proteins (including, but not limited to, β -barrels) across the mitochondrial outer membrane (Supplementary Figure 1A, step i). Our data demonstrate that darobactin A inhibits the assembly and mitochondrial membrane insertion of β -barrel proteins by the SAM complex.

The assay we used to demonstrate darobactin A inhibition of the SAM complex *in vitro* is established in the field and referred to as an “import assay”, though it captures both protein import and outer membrane insertion.

The import assay experiment protocol includes two centrifugation steps to isolate and then wash the mitochondria following the incubation with lysate containing radiolabeled substrates. Through this isolation and washing process, only the radiolabeled proteins that have been imported into the mitochondria will remain. Following solubilization of the mitochondrial membrane using detergent, we took samples before centrifugation (total) and of the supernatant following centrifugation (soluble) and ran them on an SDS-PAGE gel. A band corresponding to the radiolabeled substrate was observed on the SDS-PAGE for all substrates, time points, and darobactin A concentrations indicating that the substrate was indeed imported into the mitochondria, as any proteins not imported into the mitochondria are removed during mitochondrial isolation and subsequent wash steps (data available upon request).

Therefore, we can confirm that the import pathway remains functional, despite the inhibition of β -barrel biogenesis by darobactin A binding to the SAM complex. For this reason, we did not conduct import assays with any other mitochondrial proteins.

Does Darobactin a compete with the binding of the beta-barrel signal to the SAM complex? The authors could investigate whether Darobactin a inhibits binding of beta-barrel precursor proteins to the SAM complex. One possibility would be to import Tom40G354A variant that arrest at the SAM complex.

Yes, darobactin A competes with the binding of β -barrel proteins to the SAM complex. Our import assay experiment (Figure 3) demonstrates this, where increasing concentrations of darobactin A correlate with decreased amounts of folded β -barrel proteins. In these experiments, isolated mitochondria are incubated with darobactin A (or the same volume of water for the control sample) prior to introduction of the radiolabeled precursor protein.

The authors identified ergosterol in the structure of the SAM complex, which is surprising since ergosterol is not a major component of mitochondrial membranes. In

this context, I recommend to extend the discussion of phospholipids and their role for the SAM complex.

The *Saccharomyces cerevisiae* OMM is composed of phospholipids phosphatidylcholine (44%), phosphatidylethanolamine (34%), phosphatidylinositol (14%), cardiolipin (5%), and phosphatidylserine (4%) (Sperka-Gottlieb *et al.*, 1988, ref. 49). Both mitochondrial membranes contain sterols. The ergosterol to phospholipid ratio is higher in the IMM than in the OMM, 7.0 wt% vs 2.1 wt%, respectively (Sperka-Gottlieb *et al.*, 1988, ref. 49).

All the recombinant SAM complexes in our studies were expressed in and purified from yeast (*S. cerevisiae*) mitochondrial membranes. During this process, ergosterol was captured and enriched in our samples.

We have confirmed the presence of ergosterol in our purified SAM complex samples by LC/MS. We have added this data to the manuscript (Supplementary Figure 7), and have added a paragraph in the discussion regarding lipids and the SAM complex.

Reviewer #3 (Remarks to the Author):

The manuscript "The dynamic lateral gate of the mitochondrial β -barrel biogenesis machinery is blocked by darobactin A" by authors Diederichs *et al.* is a well-executed study on the structure and function of the mitochondrial outer membrane protein assembly machinery. The results add important new information to our understanding of the SAM complex and its inhibition, and the manuscript is very well written.

It is not completely clear to me what the authors aimed for by their molecular dynamics simulations, as most beta-barrels in their simulations seem to show a transition state between the open and closed barrels while not providing further information on the functional mechanism of SAM.

While we agree that the simulations often capture intermediate states between open and closed conformations, we believe these states are functionally relevant and reflect the inherent plasticity of the β -barrel during substrate transport or interactions with other components of the SAM complex. The aim of our molecular dynamics simulations was to investigate the conformational flexibility of the β -barrel domain of Sam50 and to capture dynamic transitions that may be relevant to its function. By simulating Sam50 both in isolation and in the presence of other SAM complex subunits (Sam35 and Sam37), we observed that these accessory subunits restrict lateral gate dynamics by preventing zipper-like closure, similar to what is observed in BamA.

Also, I find the h-bond counting a bit confusing, for example in the caption to Figure 1G "Sam50 in a partially closed conformation, observed with, at most, a couple of hydrogen bonds". Does that mean hydrogen bonds are formed during the transition from the closed to the open state? This is in contrast with statements made in the discussion, e.g. "the closed conformation of Sam50 does not have any hydrogen bonds, therefore

sampling different conformations would be more energetically favorable as no hydrogen bonds would be broken to open the gate". Please can the authors clarify?

To distinguish it from the other closed conformation, the name for the Figure 1G has been changed to "zipped closed". SAM^{op} and SAM^{cl} have no hydrogen bonds while the zipped conformation has a few hydrogen bonds. We have updated the figure caption accordingly.

Irrespective of the fact that more could have been achieved in the MD simulations in my view, the observations that lipids cannot enter at least the semi-open state of the barrel and that the gate region leads to a strong thinning of the surrounding membrane are interesting and important. In the MD methods, information about the force field used and some other important MD parameters seems to be missing, however.

We added a sentence to clarify the details in the MD methods section (now on page 37).

I also wondered, as the authors mentioned that simulations were carried out in a realistic mitochondrial membrane model, what the actual composition of the membrane was. They only provide a literature citation of the measured composition - is their model of the exact same composition?

The actual membrane composition used in the simulations is described in lines 754-757 on page 37.

Overall though, the manuscript is well-rounded and provides a wealth of interesting and new information.

Nature Communications Manuscript Submission NCOMMS-25-17724

Authors' Responses to Round 2 Reviewer Comments

Reviewer #1 (Remarks to the Author):

Thanks to the authors for thoughtfully addressing my and other reviewer's comments. The new LC-MS lipid analysis convincingly shows the presence of ergosterol in the purified SAM complex, and I have no further issues with including the ergosterol in the models. Changes to the figures and the text are appropriate and improve clarity. I recommend publication of the revised manuscript.

Thank you!

Reviewer #2 (Remarks to the Author):

The revised version of the manuscript by Diederichs and colleagues improved. However, it is not entirely clear whether incubation of mitochondria with Darobactin specifically affects the import of beta-barrel proteins and not the import of other mitochondrial proteins like presequence-containing precursor protein. Except of this point, the authors addressed my concerns. The findings are interesting and the manuscript is suitable for a broad readership of Nature Communication.

Thank you for this feedback. We have added the statement "We focused on the import and assembly of the mitochondrial proteins relevant to the SAM complex function, β -barrel proteins, and did not evaluate the import of other mitochondrial proteins such as pre-sequence containing proteins." to the relevant results section of the manuscript (page 12, lines 224-247) for further clarification.

Reviewer #3 (Remarks to the Author):

The authors have made adequate changes to their manuscript. I recommend publication.

Thank you!